# Generalized Gradient Norm Clipping
# & Non-Euclidean $(L_0, L_1)$-Smoothness

**Thomas Pethick**[*]
EPFL (LIONS)
thomas.pethick@epfl.ch

**Wanyun Xie**[*]
EPFL (LIONS)
wanyun.xie@epfl.ch

**Mete Erdogan**
EPFL (LIONS)
mete.erdogan@epfl.ch

**Kimon Antonakopoulos**
EPFL (LIONS)
kimon.antonakopoulos@epfl.ch

**Antonio Silveti-Falls**
Université Paris-Saclay (CVN)
tonys.falls@gmail.com

**Volkan Cevher**
EPFL (LIONS)
volkan.cevher@epfl.ch

## Abstract

This work introduces a hybrid non-Euclidean optimization method which generalizes gradient norm clipping by combining steepest descent and conditional gradient approaches. The method achieves the best of both worlds by establishing a descent property under a generalized notion of $(L_0, L_1)$-smoothness. Weight decay is incorporated in a principled manner by identifying a connection to the Frank-Wolfe short step. In the stochastic case, we show an order optimal $O(n^{-1/4})$ convergence rate by leveraging a momentum based gradient estimator. We discuss how to instantiate the algorithms for deep learning, which we dub Clipped Scion, and demonstrate their properties on image classification and language modeling. The code is available at https://github.com/LIONS-EPFL/ClippedScion.

## 1 Introduction

Recent work [Pethick et al., 2025] has shown that conditional gradient methods[2], traditionally used for constrained optimization, can also solve unconstrained problems—offering an alternative to steepest descent. From their analysis it becomes apparent that the two methods have distinct properties: whereas steepest descent requires the stepsize $\gamma$ for $L$-smooth objectives to be taken as $\gamma < 2/L$, conditional gradient methods have no such requirement, thus allowing for large stepsizes, while remaining stable.

The price to pay for the stability is that conditional gradient based methods are not descent methods and thus eventually needs a diminishing stepsize to converge, even in the deterministic case. The problem becomes very apparent if the iterates are close to the solution, since the iterates always move by a fixed magnitude and are thus pushed away from the solution. Steepest descent does not suffer from the same problem since the effective stepsize automatically becomes smaller as the iterates approach a solution. This observation naturally raises the following question:

---

[*]Equal contribution.

[2]By conditional gradient based methods, we mean those methods which leverage a linear minimization oracle $\mathrm{lmo}(d) = \arg\min_{x \in \mathcal{D}} \langle d, x \rangle$ when updating their parameters with an open-loop stepsize.

Table 1: Special instantiations of Algorithm 1 according to different choices of norm. Control on the norm of the parameters is guaranteed by the constrained variant of the method (Algorithm 2).

| Method | Norm type | Norm ball | lmo(d) | $\|d\|_*$ | Reference |
|---|---|---|---|---|---|
| Clipped GD | Vector | Euclidean $\|\cdot\|_2$-ball | $-\frac{d}{\|d\|_2}$ | $\|d\|_2$ | [Mikolov et al., 2012] |
| Clipped Sign | Vector | Max-norm $\|\cdot\|_\infty$-ball | $-\operatorname{sign}(d)$ | $\|d\|_1$ | **This paper** |
| Clipped Spectral | Matrix | Spectral norm $\|\cdot\|_{S_\infty}$-ball | $-UV^{\top\,1}$ | $-\operatorname{tr}(\operatorname{lmo}(d)^\top d)$ | **This paper** |
| Clipped Scion (Algorithms 3 and 4) | Product | Max-norm ball over layers | $\{r_l \operatorname{lmo}_{\|\cdot\|_{W_l}}(d_l)\}_{l\in[D]}$ | $-\sum_l \langle r_l \operatorname{lmo}(d_l), d_l \rangle$ | **This paper** |

[1] The reduced SVD is given as $d = U \operatorname{diag}(\sigma) V^\top$.

*Can we combine the two methods and get the best of both worlds? That is, does a stable method exist which takes large steps initially but adapts the stepsize when near a solution?*

In this paper we answer the above in the affirmative by considering a hybrid method that combines a conditional gradient method with steepest descent. The proposed method generalizes gradient norm clipping [Mikolov et al., 2012] beyond the Euclidean case. In practice, gradient norm clipping has been widely adopted to stabilize training of recurrent neural networks (RNNs), Transformers and diffusion models, especially in large-scale settings. Theoretically, a precise characterization of the benefits has emerged under the $(L_0, L_1)$-smoothness assumption [Zhang et al., 2019, 2020, Koloskova et al., 2023]. Expanding on this, we show that these benefits of clipping can be made compatible with non-Euclidean methods. Besides clipping, we provide a novel analysis of conditional gradient methods without clipping under these same smoothness assumptions.

Concretely, we make the following contributions:

(i) We introduce a hybrid method between a conditional gradient method and steepest descent (Algorithm 1), which in the Euclidean case recovers gradient norm clipping. The benefit of the hybrid method is made precise by showing a descent property under a generalized $(L_0, L_1)$-smoothness condition.

(ii) In the stochastic case we show an order optimal $O(n^{-1/4})$ rate by leveraging a momentum estimator. Convergence for a clipped algorithm with stochastic feedback appears to be new even in the Euclidean case.

(iii) We establish a connection between clipping and the short step from the Frank-Wolfe literature, which similarly enjoys a descent property. The connection enables us to combine clipping with weight decay in a principled manner that maintains convergence guarantees. We propose a stochastic variant of the short step (Algorithm 2) and establish a $O(n^{-1/4})$ rate.

(iv) We explicitly instantiate the algorithms for deep learning through a product norm over layers (Algorithms 3 and 4) and demonstrate their properties through experiments on image classification and language modeling.

## 2 Preliminaries

Given a continuously differentiable objective function $f : \mathcal{X} \to \mathbb{R}$, the classical gradient descent method (GD) with a stepsize $\gamma > 0$ can be written as

$$x^{k+1} = \arg\min_{x\in\mathcal{X}} \gamma\langle \nabla f(x^k), x\rangle + \tfrac{1}{2}\|x - x^k\|_2^2 = x^k - \gamma\nabla f(x^k). \tag{GD}$$

The normalized gradient descent method with radius $\rho > 0$ is, in comparison, defined as follows

$$x^{k+1} = \arg\min_{\|x-x^k\|_2\le\rho} \gamma\langle \nabla f(x^k), x\rangle = x^k + \rho \arg\min_{\|x\|_2\le 1} \gamma\langle \nabla f(x^k), x\rangle = x^k - \gamma\left[\rho\frac{\nabla f(x^k)}{\|\nabla f(x^k)\|_2}\right]. \tag{Normalized GD}$$

A hybrid variant is much more popular in practice,

$$x^{k+1} = \arg\min_{\|x-x^k\|_2\le\rho} \gamma\langle \nabla f(x^k), x\rangle + \tfrac{1}{2}\|x - x^k\|_2^2 = x^k - \gamma\min\{1, \tfrac{\rho}{\|\nabla f(x^k)\|_2}\}\nabla f(x^k), \tag{Clipped GD}$$

which we notice can be rewritten by combining GD and Normalized GD. Indeed, all three of these algorithms correspond to minimizing

$$\gamma\langle \nabla f(x^k), x\rangle + R(x)$$

for different choices of $R$. For GD, $R(x) = \frac{1}{2}\|x - x^k\|_2^2$ while for Normalized GD, $R(x) = \iota_{\rho\mathcal{D}}(x - x^k)$, the indicator function for Euclidean ball $\mathcal{D} = \{x \colon \|x\|_2 \leq 1\}$ scaled by the radius $\rho$; Clipped GD combines both by taking $R(x) = \frac{1}{2}\|x - x^k\|_2^2 + \iota_{\rho\mathcal{D}}(x - x^k)$. This results in the iterates of Clipped GD being generated by the update in Normalized GD if $\|\nabla f(x^k)\|_2$ is large, but reducing to the update in GD when $\|\nabla f(x^k)\|_2$ is small enough.

**Observation I**  Our first observation is that both GD and Normalized GD can be generalized to the non-Euclidean case. Define the sharp-operator [Nesterov, 2012, Kelner et al., 2014],

$$d^\sharp \in \arg\max_{x \in \mathcal{X}} \{\langle d, x \rangle - \tfrac{1}{2}\|x\|^2\}.$$

Then, we can write the (possibly non-Euclidean) steepest descent method (SD) as follows

$$x^{k+1} = x^k - \gamma[\nabla f(x^k)]^\sharp \tag{SD}$$

Observe that we recover GD when choosing the Euclidean $\ell_2$ norm.

Generalizing Normalized GD to non-Euclidean norms is possible by noticing that the normalization can be written in terms of the *linear minimization oracle* (lmo)

$$\mathrm{lmo}(d) \in \arg\min_{x \in \mathcal{D}} \langle d, x \rangle$$

where the constraint is a (now assumed to be non-Euclidean) norm-ball $\mathcal{D} := \{x \mid \|x\| \leq 1\}$. By choosing the $\ell_2$-norm ball, Normalized GD can be seen as an instance of the so-called unconstrained conditional gradient method (uCG) [Pethick et al., 2025],

$$x^{k+1} = x^k + \gamma\rho\,\mathrm{lmo}(\nabla f(x^k)). \tag{uCG}$$

**Observation II**  Our second central observation is that uCG can *in general* be considered a normalized version of steepest descent. This relationship follows from noticing that the sharp operator and lmo can be defined in terms of each other. Specifically, we have that

$$\mathrm{lmo}(d) = -\frac{d^\sharp}{\|d\|_*} \qquad \text{or, equivalently,} \qquad d^\sharp = -\|d\|_* \,\mathrm{lmo}(d). \tag{1}$$

In the following section we use this observation to generalize Clipped GD to the non-Euclidean case.

## 3    Method

We propose the *generalized gradient norm clipping* method (GGNC)

$$x^{k+1} = x^k - \gamma\tau_k[d^k]^\sharp \quad \text{with} \quad \tau_k := \min\{1, \tfrac{\rho}{\|d^k\|_*}\}. \tag{GGNC}$$

There is freedom in how to compute the dual norm $\|d^k\|_*$ due to the following equivalence property for the sharp operator, $\|s\|_*^2 = \|s^\sharp\|^2 = \langle s, s^\sharp \rangle$. This form is useful, e.g., in the Euclidean case where the sharp-operator is readily available, since then $[d^k]^\sharp = d^k$.

For norm choices where the lmo is more naturally available we can equivalently write GGNC as

$$x^{k+1} = x^k + \gamma\eta_k\,\mathrm{lmo}(d^k) \quad \text{with} \quad \eta_k := \min\{\rho, \|d^k\|_*\}.$$

We have that $\|d^k\|_* = -\langle d^k, \mathrm{lmo}(d^k)\rangle$ due to the definition of the dual norm and the optimality of $\mathrm{lmo}(d^k)$. So, provided that lmo has been computed, we can obtain $\|d^k\|_*$ with very little overhead. From this rewriting we also see that $\rho$ can also be interpreted as the radius of the norm-ball constraint over which we compute the lmo.

The GGNC update rule can be seen as the solution to the following optimization problem:

$$x^{k+1} \in \arg\min_{\|x - x^k\| \leq \rho} \gamma\,\langle d^k, x - x^k \rangle + \tfrac{1}{2}\|x - x^k\|^2$$

The objective is the same quadratic approximation that gives rise to SD, but the iterates are further constrained to a trust-region of radius $\rho$ in the chosen norm, as in uCG.

---
**Algorithm 1** Generalized Gradient Norm Clipping (GGNC)
---
**Input:** Horizon $n$, init. $x^1 \in \mathcal{X}$, $d^0 = 0$, momentum $\alpha_k \in (0, 1]$, stepsize $\gamma \in (0, 1)$

  1: **for** $k = 1, \ldots, n$ **do**
  2:      Sample $\xi_k \sim \mathcal{P}$
  3:      $d^k \leftarrow \alpha_k \nabla f(x^k, \xi_k) + (1 - \alpha_k) d^{k-1}$
  4:      $v^k \leftarrow -\operatorname{lmo}(d^k)$
  5:      $\eta_k \leftarrow \min\{\rho, \langle d^k, v^k \rangle\}$
  6:      $x^{k+1} \leftarrow x^k - \gamma \eta_k v^k$
  7: Choose $\bar{x}^n$ uniformly at random from $\{x^1, \ldots, x^n\}$
**Return** $\bar{x}^n$

Equivalently to step 4-6: $x^{k+1} \leftarrow x^k - \gamma \tau_k v^k$ with $\tau_k = \min\{1, \frac{\rho}{\langle d^k, v^k \rangle^{1/2}}\}$ and $v^k = [d^k]^\sharp$.

---

---
**Algorithm 2** Stochastic Short Step Conditional Gradient (S$^3$CG)
---
**Input:** Horizon $n$, init. $x^1 \in \beta\mathcal{D} = \{x \in \mathcal{X} : \|x\| \leq \beta\}$, $d^0 = 0$, momentum $\alpha_k \in (0, 1]$, stepsize $\gamma \in (0, 1]$, ball radius $\beta > 0$

  1: **for** $k = 1, \ldots, n$ **do**
  2:      Sample $\xi_k \sim \mathcal{P}$
  3:      $d^k \leftarrow \alpha_k \nabla f(x^k, \xi_k) + (1 - \alpha_k) d^{k-1}$
  4:      $v^k \leftarrow x^k - \beta \operatorname{lmo}(d^k)$
  5:      Variant 1: $\eta_k \leftarrow \min\{\rho, \frac{\langle d^k, v^k \rangle}{\|v^k\|^2}\}$
  6:      Variant 2: $\eta_k \leftarrow \min\{\rho, \frac{\langle d^k, v^k \rangle}{4\beta^2}\}$
  7:      $x^{k+1} \leftarrow x^k - \gamma \eta_k v^k$
  8: Choose $\bar{x}^n$ uniformly at random from $\{x^1, \ldots, x^n\}$
**Return** $\bar{x}^n$

---

**Stochastic case**    In the deterministic case we can simply take the direction to be $d^k = \nabla f(x^k)$. In the stochastic case, one has to proceed with more care, since $\operatorname{lmo}(d^k)$ can be biased even when $d^k$ is unbiased, due to its potential nonlinearity. With $\alpha_k \in (0, 1]$, we define the momentum based gradient estimator

$$d^k = (1 - \alpha_k) d^{k-1} + \alpha_k \nabla f(x^k, \xi_k).$$

The final algorithm involving the momentum based gradient estimator is presented in Algorithm 1.

**Weight decay & constrained problems**    Weight decay is a very popular technique, both as a regularizer to avoid overfitting and for ensuring numerical stability. A precise characterization exists for weight decay when combined with the conditional gradient based schemes like uCG, since the resulting update reduces to the classical conditional gradient method (a.k.a. Frank-Wolfe) designed for solving constrained problems [Chen et al., 2023, D'Angelo et al., 2023, Xie and Li, 2024, Pethick et al., 2025],

$$x^{k+1} = (1 - \gamma_k) x^k + \gamma_k \beta \operatorname{lmo}(\nabla f(x^k)), \tag{CG}$$

where $\beta > 0$ is the radius of norm-ball constraint and $\gamma_k > 0$ is some stepsize to be defined. The simplicial combination ensures that the iterates remain within the constraint set $\beta\mathcal{D}$ and, as a result, ensure that $\|x^k\| \leq \beta$ for all $k$.

The CG method is not necessarily a descent method. For the classical open-loop stepsize choice $\gamma_k = 2/k+2$, it is possible to step too far in the direction given by the lmo, since the stepsize does not decrease near a critical point. Naively adopting the adaptive stepsize choice from GGNC does not seem appropriate in the constrained case, since $\|d^k\|_*$ might not necessarily be zero at a solution. Instead, we will argue that the correct analog of clipping in the constrained setting corresponds to a clipped version of the Frank-Wolfe *short step*. Like GGNC, this stepsize ensures an analogous descent property.

The short step is almost an immediate consequence of the $L$-smoothness descent lemma, from which we have

$$f(x^{k+1}) \leq f(x^k) - \gamma_k \langle \nabla f(x^k), x^k - \beta \operatorname{lmo}(\nabla f(x^k)) \rangle + \gamma_k^2 \tfrac{L}{2} \|x^k - \beta \operatorname{lmo}(\nabla f(x^k))\|^2 \qquad (2)$$

$$\leq f(x^k) - \gamma_k \langle \nabla f(x^k), x^k - \beta \operatorname{lmo}(\nabla f(x^k)) \rangle + 2\gamma_k^2 L \beta^2. \qquad (3)$$

By optimizing this bound with respect to $\gamma_k$, we arrive at two variants of the short step

$$\gamma_k \stackrel{(2)}{=} \min\{1, \tfrac{\langle \nabla f(x^k), x^k - \beta \operatorname{lmo}(\nabla f(x^k)) \rangle}{L \|x^k - \beta \operatorname{lmo}(\nabla f(x^k))\|^2}\} \quad \text{or} \quad \gamma_k \stackrel{(3)}{=} \min\{1, \tfrac{\langle \nabla f(x^k), x^k - \beta \operatorname{lmo}(\nabla f(x^k)) \rangle}{4L\beta^2}\}$$

where the second variant is useful when the norm $\| \cdot \|$ is expensive to compute. What is particularly noteworthy of these stepsize choices is that they lead to descent, i.e., $f(x^{k+1}) \leq f(x^k)$, by construction. We extend these stepsize choices to the stochastic case with Algorithm 2, where we propose a slightly different parameterization given by

$$\eta_k = \min\{\rho, \tfrac{\langle d^k, x^k - \beta \operatorname{lmo}(d^k) \rangle}{\|x^k - \beta \operatorname{lmo}(d^k)\|^2}\} \quad \text{or} \quad \eta_k = \min\{\rho, \tfrac{\langle d^k, x^k - \beta \operatorname{lmo}(d^k) \rangle}{4\beta^2}\}.$$

A careful reader might have noticed the similarity between the short step in Algorithm 2 and gradient clipping in Algorithm 1. These schemes are indeed equivalent when $v^k$ is appropriately modified in Algorithm 2 to be $-\beta \operatorname{lmo}(d^k)$. This connection motivates our parameterization of the updates in Algorithm 2, which are scaled by $\beta\gamma\eta_k$, so that the following holds

$$\beta\gamma\eta_k = \beta\gamma \min\{\rho, \tfrac{-\langle d^k, \beta \operatorname{lmo}(d^k) \rangle}{\|\beta \operatorname{lmo}(d^k)\|^2}\} = \beta\gamma \min\{\rho, \tfrac{\beta\|d^k\|_*}{\|\beta \operatorname{lmo}(d^k)\|^2}\} = \beta\gamma \min\{\rho, \tfrac{\beta\|d^k\|_*}{\beta^2}\} = \gamma \min\{\rho, \|d^k\|_*\}.$$

The modified Step 7 of Algorithm 2 then becomes

$$x^{k+1} = x^k + \gamma \min\{\rho, \|d^k\|_*\} \operatorname{lmo}(d^k)$$

which is exactly what is used in GGNC.

## 3.1 Norm choices

Algorithm 1 and Algorithm 2 crucially generalize beyond the Euclidean case of Clipped GD. The following section focuses on the unconstrained variant (Algorithm 1) for simplicity, but its constrained counterpart follows in a straightforward way through Algorithm 2.

**Sign** A simple non-Euclidean example is the $\ell_\infty$ vector norm for which GGNC reduces to a *sign-based* update

$$x^{k+1} = x^k - \gamma\eta_k \operatorname{sign}(d^k) \qquad \text{(Clipped Sign)}$$

where $\eta_k := \min\{\rho, \|d^k\|_1\}$. The update is dense in the sense that each coordinate undergoes the same magnitude change.

**Spectral** The matrix analog of the $\ell_\infty$ norm is the Schatten-$\infty$ matrix norm, a.k.a. the spectral norm, which induces the following update

$$x^{k+1} = x^k - \gamma\eta_k U^k (V^k)^\top \qquad \text{(Clipped Spectral)}$$

where the reduced singular value decomposition (SVD) is given as $d^k = U^k \operatorname{diag}(\sigma^k)(V^k)^\top$. The dual norm can be computed given the lmo as $\|\sigma^k\|_1 = \|d^k\|_{\mathcal{S}_1} = -\langle d^k, \operatorname{lmo}(d^k) \rangle = -\operatorname{tr}(\operatorname{lmo}(d^k)^\top d^k) = -\operatorname{flatten}(\operatorname{lmo}(d^k))^\top \operatorname{flatten}(d^k)$, where $\| \cdot \|_{\mathcal{S}_1}$ is the Schatten-1 norm, a.k.a. the nuclear norm. This scheme is a clipped variant of the stochastic spectral descent method [Carlson et al., 2015b,a].

**Product norm** The neural networks in deep learning consist of multiple layers and it will therefore be useful to consider what we will call a *product norm*. Consider $x = (W_1, ..., W_D)$. A norm of $x$ can be composed using norms on $\{W_l\}_{l \in [D]}$:

$$\|x\| = \|(\tfrac{1}{r_1}\|W_1\|_{\mathcal{W}_1}, ..., \tfrac{1}{r_D}\|W_D\|_{\mathcal{W}_D})\|_\mathcal{X}$$

for radius parameters $r_l > 0$. Notable choices of $\| \cdot \|_\mathcal{X}$ include the $\ell_1$-norm [Flynn, 2017] and the $\ell_\infty$-norm choice made by the *modular norm* [Large et al., 2024]. Interestingly, if $\|\cdot\|_\mathcal{X}$ is the max-norm, $\| \cdot \|_\mathcal{X} = \| \cdot \|_\infty$, then:

   (i) The lmos can be computed separately as $\operatorname{lmo}_\mathcal{X}(x) = \{r_1 \operatorname{lmo}_{\mathcal{W}_1}(W_1), ..., r_D \operatorname{lmo}_{\mathcal{W}_D}(W_D)\}$

   (ii) The dual norm requires summing over all $l$ elements, i.e., $\|x\|_* = \sum_{l=1}^D \tfrac{1}{r_l} \|W_l\|_{\mathcal{W}_{l,*}}$.

As a particular example, consider the LARS optimizer [You et al., 2017], which performs normalized SGD layer-wise. The update rule can be written in terms of the lmo-based scheme uCG with the norm choice $\|x\| = \max_l \|W_l\|_F$. Writing the analog sharp-operator based scheme (i.e., SD), we see that it does *not* correspond to simply removing the normalization as for the $\ell_2$ norm. Instead, using the relationship (1), we see that the correct form for the hybrid GGNC method is

$$W_l^{k+1} = W_l^k - \gamma \min\{\rho, \sum_i \|d_i^k\|_F\} \frac{d_l^k}{\|d_l^k\|_F} \quad \forall l \in [D]$$

where $d^k = \{d_1^k, ..., d_D^k\}$ and $\gamma > 0$ is the stepsize. Through this duality, we see that while the lmo only requires local information, the dual norm computation (and consequently also the sharp-operator in SD) requires global information.

In Algorithms 3 and 4 of the appendix we specialize Algorithms 1 and 2 to the particular case where $\|\cdot\|_X$ is the max-norm. The resulting algorithms can be seen as clipped variants of the (unconstrained) Scion algorithm [Pethick et al., 2025] so we refer to them as (UNCONSTRAINED) CLIPPEDSCION.

## 4   Analysis

Why might it be useful to consider a hybrid of SD and uCG? As we will see, the convergence properties of the two methods are complementary.

One can show for SD under $L$-smoothness that

$$f(x^{k+1}) \leq f(x^k) - \gamma(1 - \gamma L/2)\|\nabla f(x^k)\|_*^2.$$

In other words, SD is a descent method in the sense that it decreases the function value $f(x^k)$ at every iteration. The price we pay for this descent is that the stepsize needs to be taken sufficiently small, specifically as $\gamma < 2/L$.

On the other hand, under the same $L$-smoothness assumption, uCG instead satisfies

$$f(x^{k+1}) \leq f(x^k) - \gamma\rho\|\nabla f(x^k)\|_* + \frac{L\gamma^2\rho^2}{2}.$$

Notice that this is not a descent method, due to the positive contribution of $\frac{L\gamma^2\rho^2}{2}$. However, there are no restrictions on the stepsize, and we can in fact show a fast rate of $O(1/k)$ for the norm of the gradient with a constant stepsize (as opposed to $O(1/\sqrt{k})$ of SD), albeit only to a neighborhood whose radius is proportional to $\gamma\rho$, as we formalize in the following result.

**Proposition 4.1.** *Suppose $f$ is $L$-smooth with respect to $\|\cdot\|_*$ and denote $f^\star = \inf_{x\in\mathcal{X}} f(x)$. Then, the iterates $\{x^k\}_{k\in\mathbb{N}^*}$ of uCG satisfy, for all $n \in \mathbb{N}^*$,*

$$\min_{1\leq k\leq n}\|\nabla f(x^k)\|_* \leq \frac{1}{n}\sum_{k=1}^n \|\nabla f(x^k)\|_* \leq \frac{f(x^1)-f^\star}{\gamma\rho n} + \frac{L\gamma\rho}{2}.$$

Recall that GGNC reduces to uCG when the gradient norm is large, so we can expect in the early phase GGNC will converge rapidly to a neighborhood of size $\frac{L\gamma\rho}{2}$. If the gradient norm is small in this region, then GGNC reduces to SD, which converges to an exact critical point even with constant stepsize and which can adapt to the loss landscape through the gradient norm.

We can make this intuition precise by analyzing these algorithms under the following generalization of $(L_0, L_1)$-smoothness to arbitrary norms.

**Assumption 4.2.** *The gradient $\nabla f$ is said to be $(L_0, L_1)$-smooth with $L_0, L_1 \in [0, \infty)$ if, for all $x, y \in \mathcal{X}$ with $\|x - y\| \leq \frac{1}{L_1}$, it holds*

$$\|\nabla f(x) - \nabla f(y)\|_* \leq (L_0 + L_1\|\nabla f(x)\|_*)\|x - y\|. \tag{4}$$

### 4.1   Deterministic case

We now proceed to generalizing Koloskova et al. [2023, Thm. 2.1] in the deterministic case. The main argument relies on establishing that GGNC (Algorithm 1) is a descent method even under the generalized $(L_0, L_1)$-smoothness assumption, which enables the scheme to converge even for a fixed, horizon-independent stepsize $\gamma$. For the remainder of the paper, we will always denote $f^\star := \inf_{x\in\mathcal{X}} f(x)$ (where it is understood this infimum is taken over $\beta\mathcal{D}$ for constrained problems) and $\Delta := f(x^1) - f^\star$.

**Theorem 4.3.** *Suppose Assumption 4.2 holds and let $n \in \mathbb{N}^*$. Consider $\{x^k\}_{1 \le k \le n}$ generated by GGNC with $d^k = \nabla f(x^k)$, and $\gamma \le 1/(L_0 + \rho L_1)$. Then, the following holds*

$$\min_{1 \le k \le n} \|\nabla f(x^k)\|_* \le \sqrt{\frac{\Delta}{\gamma n}} + \frac{2\Delta}{\gamma \rho n}.$$

*Specifically, with $\rho = \frac{L_0}{L_1}$ and $\gamma = \frac{1}{L_0}$, we have*

$$\min_{1 \le k \le n} \|\nabla f(x^k)\|_* \le \sqrt{\frac{L_0 \Delta}{n}} + \frac{2 L_1 \Delta}{n}.$$

*Remark* 4.4. Note that the condition $\|x^k - x^{k+1}\| \le 1/L_1$ of Assumption 4.2 required in the proof is always satisfied, since $\gamma \rho \le 1/L_1$ holds for any $\rho$. We note that descent can also be established for SD with an adaptive stepsize $\gamma_k = 1/L_0 + L_1\|\nabla f(x^k)\|_*$ (see e.g., Balles et al. [2020, C.2.2], which uses a definition of $(L_0, L_1)$-smoothness based on the Hessian).

In contrast with GGNC, uCG is not a descent method and requires a diminishing stepsize to converge as suggested by the following theorem. The uCG method trades off the descent property with being agnostic to the Lipschitz constant $L_0$.

**Theorem 4.5.** *Suppose Assumption 4.2 holds and let $n \in \mathbb{N}^*$. Consider $\{x^k\}_{1 \le k \le n}$ generated by uCG with $\gamma \rho < 1/2L_1$. Then, the following holds*

$$\min_{1 \le k \le n} \|\nabla f(x^k)\|_* \le \frac{2\Delta}{\gamma \rho n} + 2 L_0 \gamma \rho.$$

*Remark* 4.6. The assumption that $\gamma \rho \le 1/2L_1$ can be relaxed to $\gamma \rho < 1/L_1$ while still ensuring convergence, modulo a different constant in the convergence rate.

Let us now turn to the constrained case. The following theorem establishes a convergence rate for Algorithm 2 in the deterministic setting, i.e., with $d^k = \nabla f(x^k)$. The convergence rate is established for a quantity called the Wolfe-gap,

$$\max_{u \in \beta \mathcal{D}} \langle \nabla f(x), x - u \rangle,$$

which, when equal to 0, certifies that $x$ is a critical point for the constrained problem. It is the equivalent of the dual norm of the gradient but for constrained problems, since the gradient might not vanish at a critical point in the constrained setting. The theorem also includes an assumption that $f$ is $L$-smooth rather than $(L_0, L_1)$-smooth. Because the iterates of Algorithm 2 are guaranteed to never leave the compact set $\beta \mathcal{D}$, $L$-smoothness is implied by $(L_0, L_1)$-smoothness here.

**Theorem 4.7.** *Suppose $f$ is $L$-smooth and let $n \in \mathbb{N}^*$. Consider $\{x^k\}_{1 \le k \le n}$ generated by Algorithm 2 with $d^k = \nabla f(x^k)$, $\gamma \le \frac{1}{L}$, and $\rho \le L$ so that $\gamma \rho \le 1$. Then, for all $u \in \beta \mathcal{D}$, the following holds*

$$\min_{1 \le k \le n} \langle \nabla f(x^k), x^k - u \rangle \le 2\beta \sqrt{\frac{\Delta}{\gamma n}} + \frac{2\Delta}{\gamma \rho n}.$$

## 4.2 Stochastic case

We consider the following standard assumption about the bias and variance of the stochastic oracle.

**Assumption 4.8.** *For the stochastic gradient estimator $\nabla f(\cdot, \xi) : X \to \mathbb{R}^d$ the following holds.*

(i) *Unbiased:* $\mathbb{E}_\xi [\nabla f(x, \xi)] = \nabla f(x) \quad \forall x \in X$.

(ii) *Bounded variance:* $\mathbb{E}_\xi \left[ \|\nabla f(x, \xi) - \nabla f(x)\|_2^2 \right] \le \sigma^2 \quad \forall x \in X, \sigma \ge 0$.

In order to establish convergence in what follows, an important quantity to introduce is the error produced by the stochastic estimator $d^k$, which we denote by $\lambda^k := d^k - \nabla f(x^k)$.

We establish the following order optimal convergence guarantee for GGNC under $(L_0, L_1)$-smoothness using a momentum-based estimator. These convergence results for clipping with momentum appear to be new, even in the Euclidean case.

**Theorem 4.9.** *Suppose Assumptions 4.2 and 4.8 hold and let $n \in \mathbb{N}^*$. Consider the iterates $\{x^k\}_{1 \le k \le n}$ generated by Algorithm 1 with a constant stepsize $\gamma \le 1/L_0$ and $\gamma \rho \le 1/2L_1$. Then,*

$$\mathbb{E}[\|\nabla f(\bar{x}^n)\|_*] \le \frac{4\sqrt{\Delta}}{\sqrt{\gamma n}} + \frac{8\Delta}{\gamma \rho n} + 4\sqrt{\epsilon_n} + \frac{8\epsilon_n}{\rho}$$

where $\Delta := f(x^1) - f^\star$ and $\epsilon_n := \frac{1}{n} \sum_{k=1}^{n} O(\sqrt{\mathbb{E}[\|\lambda^k\|_2^2]} + \mathbb{E}[\|\lambda^k\|_2^2])$.

*Furthermore, assuming $f$ is $L$-smooth[3] and taking $\alpha = \frac{1}{\sqrt{n}}$, $\gamma = \frac{1}{\sqrt{n}L_0}$ and $\rho = \frac{L_0}{2n^{1/4}L_1}$ such that $\gamma\rho = \frac{1}{2n^{3/4}L_1}$ we have that*

$$\mathbb{E}[\|\nabla f(\bar{x}^n)\|_*] \le O(\tfrac{1}{n^{1/4}}).$$

*Remark* 4.10. For ease of exposition, the guarantee is presented with horizon-dependent parameter choices, but the result can be extended to an *any time* guarantee in a straightforward manner by choosing the parameters as a function of $k$ instead of $n$ and modifying the proofs accordingly.

In the constrained case, we have the following convergence guarantee for SCG with a clipped short step (Algorithm 2) using a momentum-based estimator. To the best of our knowledge, this is the first convergence proof using the short step in the stochastic setting.

**Theorem 4.11.** *Suppose Assumptions 4.2 and 4.8 hold and let $n \in \mathbb{N}^*$. Consider the iterates $\{x^k\}_{1 \le k \le n}$ generated by Algorithm 2 (Variant 1) with a constant stepsize $\gamma \le \frac{1}{L\sqrt{n}}$ and $\rho \le \frac{1}{n^{1/4}}$. Then, for all $u \in \beta\mathcal{D}$,*

$$\mathbb{E}[\langle \nabla f(\bar{x}^n), \bar{x}^n - u \rangle] \le \frac{4\sqrt{\Delta}}{\sqrt{\gamma n}} + \frac{8\Delta}{\gamma\rho n} + 4\sqrt{\epsilon_n} + \frac{8\epsilon_n}{\rho}$$

*where $\Delta := f(x^1) - f^\star$ and $\epsilon_n := \frac{1}{n} \sum_{k=1}^{n} O(\sqrt{\mathbb{E}\|\lambda^k\|_2^2} + \mathbb{E}\|\lambda^k\|_2^2)$.*

*Furthermore, taking $\alpha = \frac{1}{\sqrt{n}}$, $\gamma = \frac{1}{(L\sqrt{n})}$ and $\rho = \frac{1}{n^{1/4}}$ such that $\gamma\rho = \frac{1}{(Ln^{3/4})}$ we have that*

$$\mathbb{E}[\langle \nabla f(\bar{x}^n), \bar{x}^n - u \rangle] \le O(\tfrac{1}{n^{1/4}}).$$

We additionally provide an identical guarantee for Algorithm 2 (Variant 2) in the appendix.

## 5  Related work

$(L_0, L_1)$**-smoothness**  An $(L_0, L_1)$-smoothness condition was introduced based on the Hessian in [Zhang et al., 2019] and later generalized to the first-order notion that we extend to the non-Euclidean case [Zhang et al., 2020]. $(L_0, L_1)$-smoothness was used to analyze signSGD under heavy-tailed noise assumptions in Kornilov et al. [2025]. A coordinate-wise $(L_0, L_1)$-smoothness condition has also been considered for analyzing a generalized version of signSGD [Crawshaw et al., 2022].

In the Euclidean case, a descent property under $(L_0, L_1)$-smoothness was shown for both gradient clipping [Zhang et al., 2020, Koloskova et al., 2023] and gradient descent with an appropriate adaptive stepsize as studied in the two concurrent works Gorbunov et al. [2024] and Vankov et al. [2024].

**Parameter-agnostic**  In the deterministic case, gradient descent with backtracking line-search was shown to converge under $(L_0, L_1)$-smoothness without knowledge of the Lipschitz constants [Hübler et al., 2024]. For (star)-convex problems, an interesting connection was established between gradient norm clipping and the Polyak stepsize in Takezawa et al. [2024] and further analyzed in Gorbunov et al. [2024] and Vankov et al. [2024]. The adaptive stepsize removes the need for knowing both $L_0$ and $L_1$. Unfortunately, the Polyak stepsize is deeply tied to the Euclidean and (star)-convex structure and thus does not seem to be directly extendable to our more general setting.

In the stochastic case, the current best known parameter-agnostic method introduces an undesirable exponential dependency on $L_1$ in the complexity [Hübler et al., 2024]. However, knowledge of $L_0$ can be removed without such issues through either an AdaGrad type stepsize [Wang et al., 2023, Faw et al., 2023] or normalized gradient descent with momentum [Cutkosky and Mehta, 2020] as shown in Hübler et al. [2024]. This mirrors results from the online learning community where both AdaGrad and gradient normalization are known to adapt to Hölder smoothness [Orabona, 2023].

**Short step**  In contrast with gradient descent, the Frank-Wolfe algorithm [Frank et al., 1956] does not ensure descent with an open-loop stepsize even in the deterministic setting. Descent can be ensured by an adaptive stepsize known as the short step, originally introduced by Frank & Wolfe [Frank et al., 1956] and extended by Rubinov & Dem'yanov [Dem'yanov and Rubinov, 1968]. See Pokutta [2024] for an expository treatment.

---

[3]Only local Lipschitz is needed in the sense that the condition only needs to hold for $(x, y)$ satisfying $\|x - y\| \le \frac{1}{L_1}$. It is also possible to replace $L$ with $L_n := \max_{k \le n} L_0 + L_1 \|f(x^k)\|_*$, e.g., as is done in [Koloskova et al., 2023, Thm. 2.3].

**Spectral norm methods**  Clipped Spectral can be viewed as a hybrid method between the stochastic spectral descent [Carlson et al., 2015b] and the Muon optimizer [Jordan et al., 2024b], with some crucial differences.

Muon builds the gradient estimator $d^k$ differently. Specifically they take $d^k = \nabla f(x^k, \xi_k) + \beta d^{k-1}$ if Nesterov momentum is disabled. This is equivalent to our choice $d^k = \alpha \nabla f(x^k, \xi_k) + (1 - \alpha) d^{k-1}$ for LMO-based schemes, since the LMO is scale-invariant (i.e., $\text{lmo}(a \cdot s) = \text{lmo}(s)$ for $a > 0$) [Pethick et al., 2025]. However, for SD this equivalence no longer holds (in fact we have $[a \cdot s]^\sharp = a[s]^\sharp$ for $a \in \mathbb{R}$). The appropriate choice of $d^k$, which generalizes to SD and GGNC, turns out to be the convex combination.

Stochastic spectral descent [Carlson et al., 2015b] does not construct a gradient estimator and instead takes $d^k = \nabla f(x^k, \xi_k)$. This restricts their convergence result to the case of (mild) relative noise.

In this sense, Clipped Spectral could just as well be called Clipped Muon (not to be confused with the unrelated MuonClip [Team et al., 2025]) but we prefer Clipped Spectral as the algorithm itself is not tied to momentum nor to Newton-Schulz, as the name Muon fundamentally is.

Tuddenham et al. [2022] also studied an optimization algorithm focused on orthogonalization, however they orthogonalize *before* doing the momentum step. Pethick et al. [2025] analyzed a more general algorithm called Averaged LMO Directional Descent which admits as a special case the algorithm studied in Tuddenham et al. [2022]; their empirical and theoretical findings found this algorithm to be worse than orthogonalization *after* the momentum step, e.g., the Scion family of algorithms [Pethick et al., 2025].

We note that many works have recently analyzed the convergence behavior of algorithms using spectral LMOs like Muon and Scion, starting first with Pethick et al. [2025], Li and Hong [2025] and then Kovalev [2025], Sfyraki and Wang [2025], but always under *L*-smoothness assumptions, in contrast to this work.

**Modular norm**  [Large et al., 2024] introduced a norm choice for neural networks and established a smoothness condition for a given neural network provided the parameter remains bounded. The dual norm computation needed in GGNC is particularly easy to implement in the accompanying Modula software package since $\|d\|_* := -\text{flatten}(\text{lmo}(d))^\top \text{flatten}(d)$, which in Modula code reads as `dual_norm=-sum(model.dualize(d)*d)`.

**Weight decay**  Weight decay [Pratt, 1992] is a crucial component in deep learning and has become standard in training modern neural networks through its integration with Adam [Loshchilov and Hutter, 2017]. When combined with LMO based updates such as sign descent and the normalized gradient descent the resulting methods can be seen as instantiations of the conditional gradient method for constrained optimization problems [Chen et al., 2023, D'Angelo et al., 2023, Xie and Li, 2024, Pethick et al., 2025]. Our adaptive stepsize in Algorithm 2 effectively scales the weight decay as well as the update. This is similar to scheduled weight decay [Xie et al., 2023] which uses the adaptive stepsize in Adam to also scale the weight decay parameter.

## 6   Experiments

For the norm choice of Scion and ClippedScion we use the (Sign → Spectral → Sign) and (Spectral → Spectral → Sign) configurations for language modeling and image classication respectively (see Pethick et al. [2025, Tbl. 2-4] for the associated scaling factors). To compute the spectral lmo we use the efficient implementation provided in Jordan et al. [2024b] of the Newton-Schultz iteration proposed in Bernstein and Newhouse [2024]. There have been recent efforts to move beyond the "N" (Newton-Schulz) in Muon, the most popular algorithm computing the spectral LMO, through alternative subroutines; our algorithm is compatible with these alternatives, like the optimized PolarExpress routine [Amsel et al., 2025] or power iterations [Ahn et al., 2025, Vogels et al., 2019], although we do not explore them here.

**Image classification**  We test on a convolutional neural network (CNN) on the CIFAR10 dataset. Hyperparameters can be found in Table 2 in Appendix C. We consider both a fixed stepsize setting and stepsize scheduling using linear rampdown to investigate if the theoretical results are predictive of practice. We report the experimental results in Figure 1 where mean and standard deviation are computed over 5 independent runs.

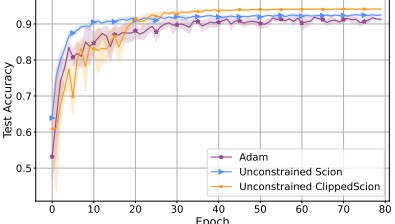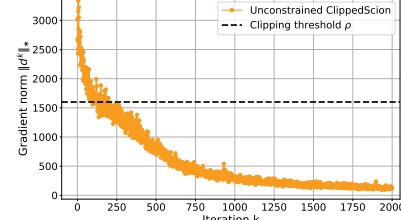

Figure 1: For CIFAR10 experiments with fixed stepsize clipping leads to a substantial improvement.

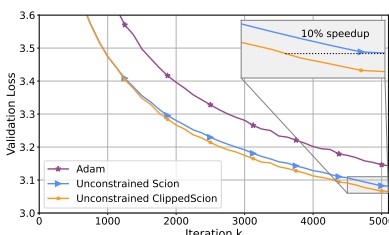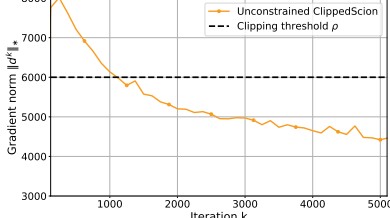

Figure 2: For fixed stepsize comparison clipping improves over Scion by more than a 10% speedup on NanoGPT (1B). We observe similar gains on the smaller 124M parameter model size (*cf.* Appendix C).

We find that clipping can substantially improve the test accuracy in the fixed stepsize setting, when the gradient norm (i.e. $\|d^k\|_* = \langle d^k, v^k \rangle$) is decreasing. This separation is in agreement with the theoretical separation between Theorem 4.3 and Theorem 4.5 on fixed stepsizes. In the constrained case (Algorithm 2) we surprisingly find that $\langle d^k, v^k \rangle$ is increasing (cf. Figure 6 in Appendix C) which requires further investigation. With stepsize scheduling we observe that clipping (i.e., Unconstrained ClippedScion) and normalization (i.e., Unconstrained Scion) achieve similar performance, which aligns with the matching theoretical rates of GGNC (Theorem 4.9) and uSCG (Pethick et al. [2025, Thm. 5.4]) in the stochastic case when stepsizes are taken decreasing.

We also evaluate the unconstrained case (Algorithm 1) using Vision Transformers (ViT) on the ImageNet dataset. We train a DeiT-base model using the DeiT codebase [Touvron et al., 2021] with replacing LayerNorm by RMS norm following [Pethick et al., 2025]. Table 3 in Appendix C contains the hyperparameter details. As shown in Figure 7 (Appendix C), Unconstrained ClippedScion achieves an 11% speedup over Unconstrained Scion, even though its gradient norm ($\|d^k\|_*$) is increasing. This observation requires further exploration.

**NanoGPT** We additionally test on NanoGPT Karpathy [2023] in Figure 2 with modernizations following [Jordan et al., 2024a]: rotary embeddings are used instead of positional embeddings, RMS norm is used instead of LayerNorm, and the ReLU$^2$ [So et al., 2021] instead of GELU activation function. All methods are trained for 5100 iterations with a batchsize of 512 and context length of 1024 on the FineWeb dataset (see Table 4 Appendix C for further details). The empirical observations matches those for CIFAR10 experiments.

## 7 Conclusion

We have shown that clipping can be extended to non-Euclidean settings and even constrained problems by establishing a precise connection to the Frank-Wolfe short step. A descent property was established under a generalized notion of $(L_0, L_1)$-smoothness, which opens up a range of interesting directions:

The descent property both in the unconstrained and constrained case enables integration with adaptive stepsize choices such as AdaGrad and backtracking line-search.

The non-Euclidean notion of $(L_0, L_1)$-smoothness we introduce might be a suitable condition to study for neural networks. Large et al. [2024] showed that neural networks are smooth in the modular norm provided that the parameters are constrained. However, in practice, violating the constraints seem to be unproblematic for optimization, which suggests that a looser smoothness assumption might hold such as Assumption 4.2.

## Acknowledgment

This work was supported as part of the Swiss AI Initiative by a grant from the Swiss National Supercomputing Centre (CSCS) under project ID a06 on Alps. This work was supported by the Swiss National Science Foundation (SNSF) under grant number 200021_205011. This work was supported by Hasler Foundation Program: Hasler Responsible AI (project number 21043). Research was sponsored by the Army Research Office and was accomplished under Grant Number W911NF-24-1-0048.

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

# Appendix

**Table of Contents**

---

**Algorithm 3** Unconstrained ClippedScion

**Input:** Horizon $n$, init. $x^1 = (W_1^1, ..., W_D^1)$, $d^0 = 0$, momentum $\alpha_k \in (0, 1]$, stepsize $\gamma \in (0, 1)$, radii $r_l \in \mathbb{R}_+$, and $\rho > 0$.

1: **for** $k = 1, \ldots, n - 1$ **do**
2:      Sample $\xi_k \sim \mathcal{P}$
3:      $d^k \leftarrow \alpha_k \nabla f(x^k, \xi_k) + (1 - \alpha_k) d^{k-1}$
4:      $v_l^k \leftarrow -r_l \operatorname{lmo}_{\|\cdot\|_{W_l}}(d_l^k) \quad \forall 1 \leq l \leq D$
5:      $\eta_k \leftarrow \min\{\rho, \sum_{l=1}^D \langle d_l^k, v_l^k \rangle\}$
6:      $x^{k+1} \leftarrow x^k - \gamma \eta_k v^k$
7: Choose $\bar{x}^n$ uniformly at random from $\{x^1, \ldots, x^n\}$

**Return** $\bar{x}^n$

---

**Algorithm 4** ClippedScion

**Input:** Horizon $n$, init. $x^1 = (W_1^1, \ldots, W_D^1) \in r_1 \mathcal{D}_1 \times \cdots \times r_D \mathcal{D}_D$, $d^0 = 0$, stepsize $\gamma \in (0, 1)$, momentum $\alpha_k \in (0, 1]$

1: **for** $k = 1, \ldots, n$ **do**
2:      Sample $\xi_k \sim \mathcal{P}$
3:      $d^k \leftarrow \alpha_k \nabla f(x^k, \xi_k) + (1 - \alpha_k) d^{k-1}$
4:      $v_l^k \leftarrow x_l^k - r_l \operatorname{lmo}_{\|\cdot\|_{W_l}}(d_l^k) \quad \forall 1 \leq l \leq D$
5:      Variant 1: $\eta_k \leftarrow \min\{\rho, \frac{\sum_{l=1}^D \langle d_l^k, v_l^k \rangle}{\max_{l=1}^D \|v_l^k\|_{W_l}^2}\}$
6:      Variant 2: $\eta_k \leftarrow \min\{\rho, \frac{\sum_{l=1}^D \langle d_l^k, v_l^k \rangle}{4}\}$
7:      $x^{k+1} \leftarrow x^k - \gamma \eta_k v^k$
8: Choose $\bar{x}^n$ uniformly at random from $\{x^1, \ldots, x^n\}$

**Return** $\bar{x}^n$

---

## A    Preliminaries

Throughout, $L$-smoothness is defined as follows.

**Definition A.1.** *A gradient mapping $\nabla f : \mathcal{X} \to \mathbb{R}^d$ is said to be L-smooth with $L \in (0, \infty)$ if for all $x, y \in \mathcal{X}$ it holds that,*

$$\|\nabla f(x) - \nabla f(y)\|_* \leq L\|x - y\|. \tag{5}$$

The sharp operator has the following properties

$$\langle s, s^\sharp \rangle = \|s^\sharp\|^2 = \|s\|_*^2 \tag{6}$$

See Kelner et al. [2014, App. A.1] for the proof.

## B    Proofs for Section 4 (Analysis)

**Proposition 4.1.** *Suppose $f$ is L-smooth with respect to $\|\cdot\|_*$ and denote $f^\star = \inf_{x \in \mathcal{X}} f(x)$. Then, the iterates $\{x^k\}_{k \in \mathbb{N}^*}$ of uCG satisfy, for all $n \in \mathbb{N}^*$,*

$$\min_{1 \leq k \leq n} \|\nabla f(x^k)\|_* \leq \frac{1}{n} \sum_{k=1}^n \|\nabla f(x^k)\|_* \leq \frac{f(x^1) - f^\star}{\gamma \rho n} + \frac{L\gamma\rho}{2}.$$

*Proof.* By the descent lemma for $L$-smooth functions applied at the points $x^k$ and $x^{k+1}$ and the definition of $x^{k+1}$ we have, for all $k \geq 1$,

$$f(x^{k+1}) \leq f(x^k) - \gamma\rho\|\nabla f(x^k)\|_* + \frac{L}{2}\gamma^2\rho^2.$$

Summing from $k = 1$ to $k = n$, and dividing by $n$ gives

$$\frac{1}{n} \sum_{k=1}^n \|\nabla f(x^k)\|_* \leq \frac{f(x^1) - f^\star}{\gamma\rho n} + \frac{L\gamma\rho}{2}.$$

Remarking that the minimum summand is smaller than the average completes the proof. $\qquad\square$

## B.1 Deterministic case

Recall the notation $\Delta := f(x^1) - f^\star$.

**Theorem 4.3.** *Suppose Assumption 4.2 holds and let $n \in \mathbb{N}^*$. Consider $\{x^k\}_{1 \le k \le n}$ generated by GGNC with $d^k = \nabla f(x^k)$, and $\gamma \le 1/(L_0 + \rho L_1)$. Then, the following holds*

$$\min_{1 \le k \le n} \|\nabla f(x^k)\|_* \le \sqrt{\frac{\Delta}{\gamma n}} + \frac{2\Delta}{\gamma \rho n}.$$

*Specifically, with $\rho = \frac{L_0}{L_1}$ and $\gamma = \frac{1}{L_0}$, we have*

$$\min_{1 \le k \le n} \|\nabla f(x^k)\|_* \le \sqrt{\frac{L_0 \Delta}{n}} + \frac{2 L_1 \Delta}{n}.$$

*Proof.* For each $1 \le k \le n$, we can write the formula for $x^{k+1}$ as follows

$$x^{k+1} = x^k - \gamma \tau_k [\nabla f(x^k)]^\sharp$$

with $\tau_k = \min\{1, \frac{\rho}{\|\nabla f(x^k)\|_*}\}$.

From $(L_0, L_1)$-smoothness and properties of the sharp-operator $\langle s, s^\sharp \rangle = \|s^\sharp\|^2 = \|s\|_*^2$, we have

$$f(x^{k+1}) \le f(x^k) + \langle \nabla f(x^k), -\gamma \tau_k \nabla f(x^k) \rangle + \frac{L_0 + \|\nabla f(x^k)\|_* L_1}{2} (\gamma \tau_k \|\nabla f(x^k)\|_*)^2$$

$$= f(x^k) - \gamma \tau_k \|\nabla f(x^k)\|_*^2 + \frac{\gamma^2 \tau_k^2}{2}(L_0 + \|\nabla f(x^k)\|_* L_1)\|\nabla f(x^k)\|_*^2.$$

A useful observation is that by definition of $\tau_k$ we have

$$\tau_k \|\nabla f(x^k)\|_* \le \rho,$$

since if $\|\nabla f(x^k)\|_* > \rho$ then $\tau_k = \rho / \|\nabla f(x^k)\|_*$, and if $\|\nabla f(x^k)\|_* \le \rho$ then $\tau_k = 1$. Thus we can upper-bound the term $\tau_k \|\nabla f(x^k)\|_* L_1$ by $\rho L_1$ in the quadratic part, yielding

$$f(x^{k+1}) \le f(x^k) - \gamma \tau_k \|\nabla f(x^k)\|_*^2 + \frac{\gamma^2 \tau_k}{2}(\tau_k L_0 + \rho L_1)\|\nabla f(x^k)\|_*^2$$

$$\le f(x^k) - \gamma \tau_k (1 - \frac{\gamma}{2}(L_0 + \rho L_1))\|\nabla f(x^k)\|_*^2$$

$$\le f(x^k) - \frac{\gamma \tau_k}{2}\|\nabla f(x^k)\|_*^2.$$

where the middle inequality uses that $\tau^2 \le \tau$ since $\tau \le 1$ and the last inequality uses the stepsize choice $\gamma \le \frac{1}{L_0 + \rho L_1}$.

There are now two cases to consider.

**Case I** Clipping Active ($\|\nabla f(x^k)\|_* > \rho$).

Here, we have $\tau_k = \frac{\rho}{\|\nabla f(x^k)\|_*}$ so

$$\tau_k \|\nabla f(x^k)\|_*^2 = \rho \|\nabla f(x^k)\|_*.$$

Therefore, the descent inequality in this case reads

$$f(x^{k+1}) \le f(x^k) - \frac{\rho \gamma}{2}\|\nabla f(x^k)\|_*.$$

**Case II** No Clipping ($\|\nabla f(x^k)\|_* \le \rho$).

In this regime, $\tau_k = 1$, so the inequality becomes

$$f(x^{k+1}) \le f(x^k) - \frac{\gamma}{2}\|\nabla f(x^k)\|_*^2.$$

This is the familiar descent guaranty for the classical steepest descent method with smooth functions.

By combining the two cases and summing over all $k = 1$ until $n$ we obtain

$$\frac{\gamma}{2}\left(\rho \sum_{k \in \mathcal{A}} \|\nabla f(x^k)\|_* + \sum_{k \notin \mathcal{A}} \|\nabla f(x^k)\|_*^2\right) \le f(x^1) - f^\star$$

where $\mathcal{A}$ is the set of indices where clipping is active (Case I). Since each sum is nonnegative, we can conclude that

$$\frac{1}{n}\sum_{k \notin \mathcal{A}} \|\nabla f(x^k)\|_*^2 \le \frac{2(f(x^1) - f^\star)}{\gamma n} \quad \text{and} \quad \frac{1}{n}\sum_{k \in \mathcal{A}} \|\nabla f(x^k)\|_* \le \frac{2(f(x^1) - f^\star)}{\gamma \rho n}. \tag{7}$$

Recall the following inequality for real numbers: for all $a, b > 0$, $a^2 \geq 2ab - b^2$. Applying this with $a = \|\nabla f(x^k)\|_*$ and $b > 0$ gives

$$\frac{1}{n}\sum_{k \notin \mathcal{A}}\|\nabla f(x^k)\|_*^2 \geq \frac{1}{n}\sum_{k \notin \mathcal{A}}(2b\|\nabla f(x^k)\|_* - b^2) = \frac{2b}{n}\left(\sum_{k \notin \mathcal{A}}\|\nabla f(x^k)\|_*\right) - \frac{b^2(n-|\mathcal{A}|)}{n}.$$

Substituting this estimate into (7) and using the fact that $|\mathcal{A}| \leq n$ gives

$$\frac{2b}{n}\left(\sum_{k \notin \mathcal{A}}\|\nabla f(x^k)\|_*\right) - \frac{b^2(n-|\mathcal{A}|)}{n} \leq \frac{2(f(x^1)-f^\star)}{\gamma n} \implies \frac{1}{n}\sum_{k \notin \mathcal{A}}\|\nabla f(x^k)\|_* \leq \frac{1}{2}\left(\frac{f(x^1)-f^\star}{b\gamma n} + b\right).$$

Then, choosing $b = \sqrt{\frac{f(x^1)-f^\star}{\gamma n}}$ simplifies the above to

$$\frac{1}{n}\sum_{k \notin \mathcal{A}}\|\nabla f(x^k)\|_* \leq \sqrt{\frac{f(x^1)-f^\star}{\gamma n}}.$$

Now, we can combine both cases to bound the sum over $1 \leq k \leq n$ as

$$\frac{1}{n}\sum_{k=1}^{n}\|\nabla f(x^k)\|_* \leq \sqrt{\frac{f(x^1)-f^\star}{\gamma n}} + \frac{2(f(x^1)-f^\star)}{\gamma \rho n}$$

Taking the minimum of the summand over $1 \leq k \leq n$ on the left hand side gives the final result. $\square$

**Theorem 4.5.** *Suppose Assumption 4.2 holds and let $n \in \mathbb{N}^*$. Consider $\{x^k\}_{1 \leq k \leq n}$ generated by uCG with $\gamma\rho < 1/2L_1$. Then, the following holds*

$$\min_{1 \leq k \leq n}\|\nabla f(x^k)\|_* \leq \frac{2\Delta}{\gamma\rho n} + 2L_0\gamma\rho.$$

*Proof.* We begin by invoking the descent lemma for $(L_0, L_1)$-smooth functions at the points $x^{k+1}$ and $x^k$, which is justified since $\|x^{k+1} - x^k\| = \gamma\rho \leq \frac{1}{2L_1}$. Then, applying the definition of $x^{k+1}$ we get, for all $1 \leq k \leq n$,

$$\begin{aligned}
f(x^{k+1}) &\leq f(x^k) + \gamma\rho\langle\nabla f(x^k), v^k\rangle + \gamma^2\rho^2(L_0 + L_1\|\nabla f(x^k)\|_*)\|v^k\|_*^2 \\
&= f(x^k) - \gamma\rho\|\nabla f(x^k)\|_* + \gamma^2\rho^2(L_0 + L_1\|\nabla f(x^k)\|_*) \\
&= f(x^k) + L_0\gamma^2\rho^2 + (L_1\gamma\rho - 1)\gamma\rho\|\nabla f(x^k)\|_*.
\end{aligned}$$

Rearranging the above yields

$$\frac{1}{2}\|\nabla f(x^k)\|_* \leq (1 - L_1\gamma\rho)\|\nabla f(x^k)\|_* \leq \frac{f(x^k)-f(x^{k+1})}{\gamma\rho} + L_0\gamma\rho$$

where we have used the assumption that $\gamma\rho \leq \frac{1}{2L_1}$ in the first inequality above. The desired claim immediately follows. $\square$

**Theorem 4.7.** *Suppose $f$ is L-smooth and let $n \in \mathbb{N}^*$. Consider $\{x^k\}_{1 \leq k \leq n}$ generated by Algorithm 2 with $d^k = \nabla f(x^k)$, $\gamma \leq \frac{1}{L}$, and $\rho \leq L$ so that $\gamma\rho \leq 1$. Then, for all $u \in \beta\mathcal{D}$, the following holds*

$$\min_{1 \leq k \leq n}\langle\nabla f(x^k), x^k - u\rangle \leq 2\beta\sqrt{\frac{\Delta}{\gamma n}} + \frac{2\Delta}{\gamma\rho n}.$$

*Proof.* We start by applying the descent lemma for $L$-smooth functions at the points $x^{k+1}$ and $x^k$ to get, for all $1 \leq k \leq n$,

$$f(x^{k+1}) \leq f(x^k) - \gamma\eta_k\langle\nabla f(x^k), v^k\rangle + \frac{L}{2}\gamma^2\eta_k^2\|v^k\|^2.$$

Now, we divide the analysis into two cases depending on whether or not clipping is active.

**Case I** Clipping active ($\frac{\langle \nabla f(x^k), v^k \rangle}{\|v^k\|^2} \geq \rho$; $\eta_k = \rho$).

In this case, we can use the fact that $\langle \nabla f(x^k), v^k \rangle \geq \rho \|v^k\|^2$ to get

$$
\begin{aligned}
f(x^{k+1}) &\leq f(x^k) - \gamma \eta_k \langle \nabla f(x^k), v^k \rangle + \frac{L}{2} \gamma^2 \eta_k^2 \|v^k\|^2 \\
&= f(x^k) - \gamma \rho \langle \nabla f(x^k), v^k \rangle + \frac{L}{2} \gamma^2 \rho^2 \|v^k\|^2 \\
&\leq f(x^k) - \gamma \rho \langle \nabla f(x^k), v^k \rangle + \frac{L}{2} \gamma^2 \rho \langle \nabla f(x^k), v^k \rangle \\
&\leq f(x^k) - \frac{1}{2} \gamma \rho \langle \nabla f(x^k), v^k \rangle
\end{aligned}
$$

where the final inequality is due to the assumption that $\gamma \leq \frac{1}{L}$. Rearranging this gives

$$
\frac{\gamma \rho}{2} \langle \nabla f(x^k), v^k \rangle \leq f(x^k) - f(x^{k+1}).
$$

**Case II** No clipping ($\frac{\langle \nabla f(x^k), v^k \rangle}{\|v^k\|^2} \leq \rho$; $\eta_k = \frac{\langle \nabla f(x^k), v^k \rangle}{\|v^k\|^2}$).

When clipping is not active, $\eta_k$ acts like a short step which gives

$$
\begin{aligned}
f(x^{k+1}) &\leq f(x^k) - \gamma \eta_k \langle \nabla f(x^k), v^k \rangle + \frac{L}{2} \gamma^2 \eta_k^2 \|v^k\|^2 \\
&\leq f(x^k) - \gamma \frac{\langle \nabla f(x^k), v^k \rangle^2}{\|v^k\|^2} + \frac{L}{2} \gamma^2 \frac{\langle \nabla f(x^k), v^k \rangle^2}{\|v^k\|^2} \\
&\leq f(x^k) - \gamma \frac{\langle \nabla f(x^k), v^k \rangle^2}{\|v^k\|^2}
\end{aligned}
$$

where the last inequality follows from the assumption that $\gamma \leq \frac{1}{L}$. Rearranging this gives

$$
\frac{\gamma}{4\beta^2} \langle \nabla f(x^k), v^k \rangle^2 \leq f(x^k) - f(x^{k+1}).
$$

**Combining both cases** Denoting $\mathcal{A}$ the set of indices where clipping is active and summing from $k = 1$ to $n$ we find

$$
\sum_{k \in \mathcal{A}} \frac{\gamma \rho}{2} \langle \nabla f(x^k), v^k \rangle + \sum_{k \notin \mathcal{A}} \frac{\gamma}{4\beta^2} \langle \nabla f(x^k), v^k \rangle^2 \leq f(x^1) - f^\star
$$

which, since each summand is nonnegative, implies that

$$
\frac{1}{n} \sum_{k \in \mathcal{A}} \langle \nabla f(x^k), v^k \rangle \leq \frac{2(f(x^1) - f^\star)}{\gamma \rho n} \quad \text{and} \quad \frac{1}{n} \sum_{k \notin \mathcal{A}} \langle \nabla f(x^k), v^k \rangle^2 \leq \frac{4\beta^2(f(x^1) - f^\star)}{\gamma n}. \tag{8}
$$

Recall the following inequality for real numbers: for all $a, b > 0$, $a^2 \geq 2ab - b^2$. Applying this with $a = \langle \nabla f(x^k), v^k \rangle$ and $b > 0$ gives

$$
\frac{1}{n} \sum_{k \notin \mathcal{A}} \langle \nabla f(x^k), v^k \rangle^2 \geq \frac{1}{n} \sum_{k \notin \mathcal{A}} (2b \langle \nabla f(x^k), v^k \rangle - b^2) = \frac{2b}{n} \left( \sum_{k \notin \mathcal{A}} \langle \nabla f(x^k), v^k \rangle \right) - \frac{b^2(n - |\mathcal{A}|)}{n}.
$$

Substituting this estimate into (8) and using the fact that $|\mathcal{A}| \leq n$ gives

$$
\frac{2b}{n} \left( \sum_{k \notin \mathcal{A}} \langle \nabla f(x^k), v^k \rangle \right) - \frac{b^2(n - |\mathcal{A}|)}{n} \leq \frac{4\beta^2(f(x^1) - f^\star)}{\gamma n}
$$

which implies that

$$
\frac{1}{n} \sum_{k \notin \mathcal{A}} \langle \nabla f(x^k), v^k \rangle \leq \frac{1}{2} \left( \frac{4\beta^2(f(x^1) - f^\star)}{b \gamma n} + b \right).
$$

Thus, choosing $b = \sqrt{\frac{4\beta^2(f(x^1) - f^\star)}{\gamma n}}$ simplifies the above to

$$\frac{1}{n} \sum_{k \notin \mathcal{A}} \langle \nabla f(x^k), v^k \rangle \leq \sqrt{\frac{4\beta^2(f(x^1) - f^\star)}{\gamma n}} = 2\beta \sqrt{\frac{f(x^1) - f^\star}{\gamma n}}$$

Now, we can combine both cases to bound the sum over $1 \leq k \leq n$ as

$$\frac{1}{n} \sum_{k=1}^{n} \langle \nabla f(x^k), v^k \rangle \leq 2\beta \sqrt{\frac{f(x^1) - f^\star}{\gamma n}} + \frac{2(f(x^1) - f^\star)}{\gamma \rho n}$$

Finally, by lower bounding the left hand side by the minimal summand over $1 \leq k \leq n$ and using the definition of $v^k$ we arrive, for all $u \in \beta \mathcal{D}$, at

$$\min_{1 \leq k \leq n} \langle \nabla f(x^k), x^k - u \rangle \leq 2 \left( \beta \frac{\sqrt{f(x^1) - f^\star}}{\sqrt{\gamma n}} + \frac{f(x^1) - f^\star}{\gamma \rho n} \right).$$

$\square$

## B.2 Stochastic case

### B.2.1 Convergence Analysis of uSCG

We now generalize the error control lemma Mokhtari et al. [2020, Lem. 6] to the $(L_0, L_1)$-smooth case and modify it for the clipped algorithm Algorithm 1.

**Lemma B.1** (Linear recursive inequality for $\mathbb{E} \left\| \lambda^k \right\|_2^2$ for GGNC). *Suppose Assumptions 4.2 and 4.8 hold and let $n \in \mathbb{N}^*$. Consider the iterates $\{x^k\}_{1 \leq k \leq n}$ generated by Algorithm 1. Then, for all $k \in \{1, \ldots, n\}$, it holds*

$$\mathbb{E}[\left\| \lambda^k \right\|_2^2] \leq \left( 1 - \frac{\alpha_k}{2} \right) \mathbb{E}[\left\| \lambda^{k-1} \right\|_2^2] + \alpha_k^2 \sigma^2 + \frac{4\gamma^2 \zeta_*^2 \rho^2 L_0^2}{\alpha_k} + \frac{4\gamma^2 \zeta_*^2 \rho^2 L_1^2}{\alpha_k} \|\nabla f(x^k)\|_*^2,$$

*where $\zeta_* := \max_{x \in \mathcal{X}} \frac{\|x\|_2}{\|x\|_*}$.*

*Proof.* The proof is a straightforward adaptation of the arguments laid out in Mokhtari et al. [2020, Lem. 6], which in fact do not depend on convexity nor on the choice of stepsize. Let $n \in \mathbb{N}^*$ and $k \in \{2, \ldots, n\}$, then

$$\begin{aligned}
\left\| \lambda^k \right\|_2^2 &= \left\| \nabla f(x^k) - d^k \right\|_2^2 \\
&= \left\| \nabla f(x^k) - \alpha_k \nabla f(x^k, \xi_k) - (1 - \alpha_k) d^{k-1} \right\|_2^2 \\
&= \left\| \alpha_k \left( \nabla f(x^k) - \nabla f(x^k, \xi_k) \right) + (1 - \alpha_k) \left( \nabla f(x^k) - \nabla f(x^{k-1}) \right) - (1 - \alpha_k) \left( d^{k-1} - \nabla f(x^{k-1}) \right) \right\|_2^2 \\
&= \alpha_k^2 \left\| \nabla f(x^k) - \nabla f(x^k, \xi_k) \right\|_2^2 + (1 - \alpha_k)^2 \left\| \nabla f(x^k) - \nabla f(x^{k-1}) \right\|_2^2 \\
&\quad + (1 - \alpha_k)^2 \left\| \nabla f(x^{k-1}) - d^{k-1} \right\|_2^2 \\
&\quad + 2\alpha_k(1 - \alpha_k) \langle \nabla f(x^{k-1}) - \nabla f(x^{k-1}, \xi_{k-1}), \nabla f(x^k) - \nabla f(x^{k-1}) \rangle \\
&\quad + 2\alpha_k(1 - \alpha_k) \langle \nabla f(x^k) - \nabla f(x^k, \xi_k), \nabla f(x^{k-1}) - d^{k-1} \rangle \\
&\quad + 2(1 - \alpha_k)^2 \langle \nabla f(x^k) - \nabla f(x^{k-1}), \nabla f(x^{k-1}) - d^{k-1} \rangle.
\end{aligned}$$

Taking the expectation conditioned on the filtration $\mathcal{F}_k$ generated by the iterates until $k$, i.e., the sigma algebra generated by $\{x^1, \ldots, x^k\}$, which we denote using $\mathbb{E}_k[\cdot]$, and using the unbiased property in Assumption 4.8, we get,

$$\begin{aligned}
\mathbb{E}_k[\left\| \lambda^k \right\|_2^2] &= \alpha_k^2 \mathbb{E}_k[\left\| \nabla f(x^k) - \nabla f(x^k, \xi_k) \right\|_2^2] + (1 - \alpha_k)^2 \left\| \nabla f(x^k) - \nabla f(x^{k-1}) \right\|_2^2 \\
&\quad + (1 - \alpha_k)^2 \left\| \lambda^{k-1} \right\|_2^2 + 2(1 - \alpha_k)^2 \langle \nabla f(x^k) - \nabla f(x^{k-1}), \lambda^{k-1} \rangle.
\end{aligned}$$

For brevity define $L_k := L_0 + L_1 \|\nabla f(x^k)\|_*$. From the above expression we can estimate,

$$
\begin{aligned}
\mathbb{E}_k[\|\lambda^k\|_2^2] &\overset{(a)}{\leq} \alpha_k^2 \sigma^2 + (1-\alpha_k)^2 \|\nabla f(x^k) - \nabla f(x^{k-1})\|_2^2 + (1-\alpha_k)^2 \|\lambda^{k-1}\|_2^2 \\
&\qquad + 2(1-\alpha_k)^2 \langle \nabla f(x^k) - \nabla f(x^{k-1}), \lambda^{k-1} \rangle \\
&\overset{(b)}{\leq} \alpha_k^2 \sigma^2 + (1-\alpha_k)^2 \|\nabla f(x^k) - \nabla f(x^{k-1})\|_2^2 + (1-\alpha_k)^2 \|\lambda^{k-1}\|_2^2 \\
&\qquad + (1-\alpha_k)^2 \left( \frac{\alpha_k}{2} \|\nabla f(x^k) - \nabla f(x^{k-1})\|_2^2 + \frac{2}{\alpha_k} \|\lambda^{k-1}\|_2^2 \right) \\
&\overset{(c)}{\leq} \alpha_k^2 \sigma^2 + (1-\alpha_k)^2 \zeta_*^2 \|\nabla f(x^k) - \nabla f(x^{k-1})\|^2 + (1-\alpha_k)^2 \|\lambda^{k-1}\|_2^2 \\
&\qquad + (1-\alpha_k)^2 \left( \frac{\alpha_k}{2} \zeta_*^2 \|\nabla f(x^k) - \nabla f(x^{k-1})\|^2 + \frac{2}{\alpha_k} \|\lambda^{k-1}\|_2^2 \right) \\
&\overset{(d)}{\leq} \alpha_k^2 \sigma^2 + (1-\alpha_k)^2 \zeta_*^2 L_k^2 \|x^k - x^{k-1}\|^2 + (1-\alpha_k)^2 \|\lambda^{k-1}\|_2^2 \\
&\qquad + (1-\alpha_k)^2 \left( (\tfrac{\alpha_k}{2}) \zeta_* L_k^2 \|x^k - x^{k-1}\|^2 + \frac{2}{\alpha_k} \|\lambda^{k-1}\|_2^2 \right) \\
&\overset{(e)}{\leq} \alpha_k^2 \sigma^2 + (1-\alpha_k)^2 L_k^2 \zeta_*^2 \gamma^2 \eta_k^2 + (1-\alpha_k)^2 \|\lambda^{k-1}\|_2^2 + (1-\alpha_k)^2 \left( (\tfrac{\alpha_k}{2}) L_k^2 \zeta_*^2 \gamma^2 \eta_k^2 + \frac{2}{\alpha_k} \|\lambda^{k-1}\|_2^2 \right) \\
&\overset{(f)}{\leq} \alpha_k^2 \sigma^2 + (1 + \tfrac{\alpha_k}{2})(1-\alpha_k) \zeta_*^2 L_k^2 \gamma^2 \eta_k^2 + (1 + \tfrac{2}{\alpha_k})(1-\alpha_k) \|\lambda^{k-1}\|_2^2 \\
&\overset{(g)}{\leq} \alpha_k^2 \sigma^2 + 2(1 + \tfrac{\alpha_k}{2})(1-\alpha_k) \zeta_*^2 \gamma^2 \rho^2 (L_0^2 + L_1^2 \|\nabla f(x^k)\|_*^2) + (1 + \tfrac{2}{\alpha_k})(1-\alpha_k) \|\lambda^{k-1}\|_2^2,
\end{aligned}
$$

using the bounded variance property from Assumption 4.8 for (a), Young's inequality with parameter $\alpha_k/2 > 0$ for (b), $\zeta_* := \max_{x \in \mathcal{X}} \frac{\|x\|_2}{\|x\|_*}$ for (c), Assumption 4.2 for (d), the definition of $x^k$ from Algorithm 1 for (e), the fact that $1 - \alpha_k < 1$ for (f), and $\eta_k \leq \rho$ and Young's inequality on $L_k^2$ for (g). To complete the proof, we note that, for all $k \in \{1, \dots, n\}$, it holds

$$
(1 + \tfrac{2}{\alpha_k})(1-\alpha_k) \leq \tfrac{2}{\alpha_k} \quad \text{and} \quad (1-\alpha_k)(1 + \tfrac{\alpha_k}{2}) \leq 1 - \tfrac{\alpha_k}{2}
$$

which, applied to the previous inequality and taking total expectations, yields

$$
\mathbb{E}[\|\lambda^k\|_2^2] \leq \left(1 - \frac{\alpha_k}{2}\right) \mathbb{E}[\|\lambda^{k-1}\|_2^2] + \alpha_k^2 \sigma^2 + \frac{4 \gamma^2 \zeta_*^2 \rho^2}{\alpha_k} (L_0^2 + L_1^2 \|\nabla f(x^k)\|_*^2).
$$

$\square$

**Lemma B.2** (Bound on $\mathbb{E}\|\lambda^k\|_2^2$ with horizon-dependent $\alpha$ for GGNC). *Suppose Assumptions 4.2 and 4.8 hold, $f$ is $L$-smooth, and let $n \in \mathbb{N}^*$. Consider the iterates $\{x^k\}_{1 \leq k \leq n}$ generated by Algorithm 1 with a stepsize $\gamma \rho$ satisfying*

$$
\gamma \rho < \frac{1}{2 n^{3/4} L_1}. \tag{9}
$$

*Moreover, consider a momentum $\alpha_k = \alpha = \frac{1}{\sqrt{n}}$ for all $k \in \{1, \dots, n\}$. Then, for all $k \in \{1, \dots, n\}$ the following holds*

$$
\mathbb{E}[\|\lambda^k\|_2^2] \leq \frac{2(\sigma^2 + \zeta_*^2 L^2 / L_1^2)}{\sqrt{n}}. \tag{10}
$$

*Proof.* Let $k \in \{1, \dots, n\}$. We start from the recursive inequality obtained in Lemma B.1 for $\mathbb{E}[\|\lambda^k\|_2^2]$. Since we are now assuming that $f$ is $L$-smooth, this inequality is satisfied with $L_0 = L$ and $L_1 = 0$, which gives

$$
\mathbb{E}[\|\lambda^k\|_2^2] \leq \left(1 - \frac{\alpha}{2}\right) \mathbb{E}[\|\lambda^{k-1}\|_2^2] + \alpha^2 \sigma^2 + \frac{4 \gamma^2 \rho^2 \zeta_*^2 L^2}{\alpha}. \tag{11}
$$

Now, we substitute the specific choice $\alpha = \frac{1}{\sqrt{n}}$ of momentum to find

$$
\mathbb{E}[\|\lambda^k\|_2^2] \leq (1 - \tfrac{1}{2\sqrt{n}}) \mathbb{E}[\|\lambda^{k-1}\|_2^2] + \frac{\sigma^2}{n} + 4 \zeta_*^2 L^2 \rho^2 \gamma^2 \sqrt{n}. \tag{12}
$$

Using the particular choice of $\gamma \rho$, we have

$$
\mathbb{E}[\|\lambda^k\|_2^2] \leq (1 - \tfrac{1}{2\sqrt{n}}) \mathbb{E}[\|\lambda^{k-1}\|_2^2] + \tfrac{1}{n}(\sigma^2 + \tfrac{\zeta_*^2 L^2}{L_1^2}).
$$

Let $u_k = \mathbb{E}[\|\lambda^k\|_2^2]$, $a = \frac{1}{2\sqrt{n}}$, and $b = \frac{1}{n}(\sigma^2 + \frac{\zeta_*^2 L^2}{L_1^2})$. Unrolling the recurrence relation $u_k \le (1-a)u_{k-1} + b$, we have

$$u_k \le (1-a)^k u_0 + b \sum_{i=0}^{k-1}(1-a)^i = b \sum_{i=0}^{k-1}(1-a)^i = b\frac{1-(1-a)^k}{1-(1-a)} = b\frac{1-(1-a)^k}{a}.$$

Since $0 < a < 1$, we have $0 < (1-a)^k < 1$ for $k \ge 1$. Thus, $1 - (1-a)^k < 1$. Therefore,

$$u_k \le {}^b\!/_a. \tag{13}$$

Substituting the values for $a$ and $b$, we have

$$\mathbb{E}[\|\lambda^k\|_2^2] \le \frac{2(\sigma^2 + \zeta_*^2 L^2/L_1^2)}{\sqrt{n}}. \tag{14}$$

This concludes the proof. $\qquad\square$

**Theorem 4.9.** *Suppose Assumptions 4.2 and 4.8 hold and let $n \in \mathbb{N}^*$. Consider the iterates $\{x^k\}_{1 \le k \le n}$ generated by Algorithm 1 with a constant stepsize $\gamma \le {}^1\!/_{L_0}$ and $\gamma\rho \le {}^1\!/_{2L_1}$. Then,*

$$\mathbb{E}[\|\nabla f(\bar{x}^n)\|_*] \le \frac{4\sqrt{\Delta}}{\sqrt{\gamma n}} + \frac{8\Delta}{\gamma\rho n} + 4\sqrt{\epsilon_n} + \frac{8\epsilon_n}{\rho}$$

*where $\Delta := f(x^1) - f^\star$ and $\epsilon_n := \frac{1}{n}\sum_{k=1}^n O\left(\sqrt{\mathbb{E}[\|\lambda^k\|_2^2]} + \mathbb{E}[\|\lambda^k\|_2^2]\right)$.*

*Furthermore, assuming $f$ is $L$-smooth[4] and taking $\alpha = {}^1\!/\sqrt{n}$, $\gamma = {}^1\!/\sqrt{n}L_0$ and $\rho = {}^{L_0}\!/_{2n^{1/4}L_1}$ such that $\gamma\rho = {}^1\!/_{2n^{3/4}L_1}$ we have that*

$$\mathbb{E}[\|\nabla f(\bar{x}^n)\|_*] \le O\left(\frac{1}{n^{1/4}}\right).$$

*Proof.* Given that $\|x^k - x^{k+1}\| \le {}^1\!/_{L_1}$, which we will ensure by choice of the stepsize $\gamma\eta_k$ and radius $\rho$, we have from Assumption 4.2 that

$$
\begin{aligned}
0 &\le f(x^k) - f(x^{k+1}) + \langle\nabla f(x^k), x^{k+1} - x^k\rangle + \frac{L_0 + L_1\|\nabla f(x^k)\|_*}{2}\|x^{k+1} - x^k\|^2 \\
&= f(x^k) - f(x^{k+1}) + \gamma\eta_k\langle\nabla f(x^k), \mathrm{lmo}(d^k)\rangle + \frac{L_0}{2}\gamma^2\eta_k^2\|\mathrm{lmo}(d^k)\|^2 + \frac{L_1}{2}\gamma^2\eta_k^2\|\nabla f(x^k)\|_*\|\mathrm{lmo}(d^k)\|^2 \\
&= f(x^k) - f(x^{k+1}) + \gamma\eta_k\langle\nabla f(x^k), \mathrm{lmo}(d^k)\rangle + \frac{L_0}{2}\gamma^2\eta_k^2 + \frac{L_1}{2}\gamma^2\eta_k^2\|\nabla f(x^k)\|_*
\end{aligned}
$$

where we recall that $\eta_k = \min\{\rho, \|d^k\|_*\}$.

To treat the inner product we introduce the error $\lambda^k := d^k - \nabla f(x^k)$ and proceed as follows

$$
\begin{aligned}
\langle\nabla f(x^k), \mathrm{lmo}(d^k)\rangle &= \langle\nabla f(x^k) - d^k, \mathrm{lmo}(d^k)\rangle + \langle d^k, \mathrm{lmo}(d^k)\rangle \\
&\le \langle\nabla f(x^k) - d^k, \mathrm{lmo}(d^k)\rangle - \|d^k\|_* \\
&= \langle\nabla f(x^k) - d^k, \mathrm{lmo}(d^k)\rangle - \tfrac{1}{2}\|d^k\|_* - \tfrac{1}{2}\|d^k\|_* \\
\text{(Triangle ineq.)} \quad &\le \langle\nabla f(x^k) - d^k, \mathrm{lmo}(d^k)\rangle - \tfrac{1}{2}\|d^k\|_* - \tfrac{1}{2}\|\nabla f(x^k)\|_* + \tfrac{1}{2}\|\lambda^k\|_* \\
\text{(Cauchy-Schwarz)} \quad &\le \|\lambda^k\|_* - \tfrac{1}{2}\|d^k\|_* - \tfrac{1}{2}\|\nabla f(x^k)\|_* + \tfrac{1}{2}\|\lambda^k\|_* \\
&\le \tfrac{3}{2}\zeta\|\lambda^k\|_2 - \tfrac{1}{2}\|d^k\|_* - \tfrac{1}{2}\|\nabla f(x^k)\|_*
\end{aligned}
$$

where $\zeta := \max_{x\in\mathcal{X}}\frac{\|x\|_*}{\|x\|_2}$ is the norm equivalence constant.

Combining the two inequalities we have

$$
\begin{aligned}
0 &\le f(x^k) - f(x^{k+1}) + \gamma\eta_k\tfrac{3}{2}\zeta\|\lambda^k\|_2 - \gamma\eta_k\tfrac{1}{2}\|d^k\|_* - \gamma\eta_k\tfrac{1}{2}\|\nabla f(x^k)\|_* + \frac{L_0}{2}\gamma^2\eta_k^2 + \frac{L_1}{2}\gamma^2\eta_k^2\|\nabla f(x^k)\|_* \\
&= f(x^k) - f(x^{k+1}) + \gamma\eta_k\tfrac{3}{2}\zeta\|\lambda^k\|_2 - \gamma\eta_k\tfrac{1}{2}(\|d^k\|_* - L_0\gamma\eta_k) - \gamma\eta_k\tfrac{1}{2}(1 - L_1\gamma\eta_k)\|\nabla f(x^k)\|_*
\end{aligned}
$$

**Case I** Clipping Active ($\rho < \|d^k\|_*$).

In this case we have that $\eta_k = \rho$, so

$$
\begin{aligned}
0 &\le f(x^k) - f(x^{k+1}) + \gamma_k\rho\tfrac{3}{2}\zeta\|\lambda^k\|_2 - \gamma_k\eta_k^2\tfrac{1}{2}(1 - L_0\gamma_k) - \gamma_k\rho\tfrac{1}{2}(1 - L_1\gamma_k\rho)\|\nabla f(x^k)\|_* \\
&\le f(x^k) - f(x^{k+1}) + \gamma_k\rho\tfrac{3}{2}\zeta\|\lambda^k\|_2 - \gamma_k\rho\tfrac{1}{2}(1 - L_1\gamma_k\rho)\|\nabla f(x^k)\|_*
\end{aligned}
$$

where we have used $\gamma_k \le \frac{1}{L_0}$.

---

[4]Only local Lipschitz is needed in the sense that the condition only needs to hold for $(x, y)$ satisfying $\|x - y\| \le {}^1\!/_{L_1}$. It is also possible to replace $L$ with $L_n := \max_{k \le n} L_0 + L_1\|f(x^k)\|_*$, e.g., as is done in [Koloskova et al., 2023, Thm. 2.3].

**Case II**  No Clipping $(\rho \geq \|d^k\|_*)$.

Here we have that $\eta_k = \|d^k\|_*$, so

$$0 \leq f(x^k) - f(x^{k+1}) + \gamma_k \rho \tfrac{3}{2} \zeta \|\lambda^k\|_2 - \gamma_k \eta_k^2 \tfrac{1}{2}(1 - L_0 \gamma_k) - \gamma_k \tfrac{1}{2}(1 - L_1 \gamma_k \rho) \|d^k\|_* \|\nabla f(x^k)\|_*$$

Focusing on the last term, we have

$$\|d^k\|_* \|\nabla f(x^k)\|_* = \|d^k - \nabla f(x^k) + \nabla f(x^k)\|_* \|\nabla f(x^k)\|_*$$
$$\text{(Triangle ineq.)} \geq (\|\nabla f(x^k)\|_* - \|\lambda^k\|_*) \|\nabla f(x^k)\|_*$$
$$= \|\nabla f(x^k)\|_*^2 - \|\lambda^k\|_* \|\nabla f(x^k)\|_*.$$

For the last term, using the triangle inequality, we have

$$\|\lambda^k\|_* \|\nabla f(x^k)\|_* \leq \|\lambda^k\|_* (\|\nabla f(x^k) - d^k\|_* + \|d^k\|_*)$$
$$= \|\lambda^k\|_*^2 + \|\lambda^k\|_* \|d^k\|_*$$
$$\leq \|\lambda^k\|_*^2 + \|\lambda^k\|_* \rho$$
$$\leq \zeta \|\lambda^k\|_2^2 + \zeta \|\lambda^k\|_2 \rho.$$

By combining, we have

$$0 \leq f(x^k) - f(x^{k+1}) - \gamma_k \eta_k^2 \tfrac{1}{2}(1 - L_0 \gamma_k) - \gamma_k \tfrac{1}{2}(1 - L_1 \gamma_k \rho) \|\nabla f(x^k)\|_*^2$$
$$+ \gamma_k \rho \tfrac{3}{2} \zeta \|\lambda^k\|_2 + \gamma_k \tfrac{1}{2} \zeta (1 - L_1 \gamma_k \rho)(\|\lambda^k\|_2^2 + \|\lambda^k\|_2 \rho)$$
$$= f(x^k) - f(x^{k+1}) - \gamma_k \eta_k^2 \tfrac{1}{2}(1 - L_0 \gamma_k) - \gamma_k \tfrac{1}{2}(1 - L_1 \gamma_k \rho) \|\nabla f(x^k)\|_*^2$$
$$+ \gamma_k \rho \zeta (2 - \tfrac{1}{2} L_1 \gamma_k \rho) \|\lambda^k\|_2 + \gamma_k \tfrac{1}{2} \zeta (1 - L_1 \gamma_k \rho) \|\lambda^k\|_2^2$$
$$\leq f(x^k) - f(x^{k+1}) - \gamma_k \tfrac{1}{2}(1 - L_1 \gamma_k \rho) \|\nabla f(x^k)\|_*^2$$
$$+ \gamma_k \rho \zeta (2 - \tfrac{1}{2} L_1 \gamma_k \rho) \|\lambda^k\|_2 + \gamma_k \tfrac{1}{2} \zeta (1 - L_1 \gamma_k \rho) \|\lambda^k\|_2^2$$

where the last inequality uses $\gamma_k \leq \frac{1}{L_0}$.

**Combining both cases**  Introducing the set of iterates where clipping is active, $\mathcal{A} := \{k \in [n] \mid \rho < \|d^k\|_*\}$, we can take the expectation of both sides and sum the two cases to find

$$\tfrac{1}{2} \gamma (1 - L_1 \gamma \rho)(\rho \sum_{k \in \mathcal{A}} \mathbb{E}[\|\nabla f(x^k)\|_*] + \sum_{k \notin \mathcal{A}} \mathbb{E}[\|\nabla f(x^k)\|_*^2])$$
$$\leq f(x^1) - f^\star + \sum_{k \in \mathcal{A}} \gamma \rho \zeta \tfrac{3}{2} \mathbb{E}[\|\lambda^k\|_2] + \sum_{k \notin \mathcal{A}} \gamma \rho \zeta (2 - \tfrac{1}{2} L_1 \gamma \rho) \mathbb{E}[\|\lambda^k\|_2] + \gamma \tfrac{1}{2} \zeta (1 - L_1 \gamma \rho) \mathbb{E}[\|\lambda^k\|_2^2]$$
$$\leq f(x^1) - f^\star + \sum_{k \in \mathcal{A}} \gamma \rho \zeta \tfrac{3}{2} \sqrt{\mathbb{E}[\|\lambda^k\|_2^2]} + \sum_{k \notin \mathcal{A}} \gamma \rho \zeta (2 - \tfrac{1}{2} L_1 \gamma \rho) \sqrt{\mathbb{E}[\|\lambda^k\|_2^2]} + \gamma \tfrac{1}{2} \zeta (1 - L_1 \gamma \rho) \mathbb{E}[\|\lambda^k\|_2^2]$$
$$\leq f(x^1) - f^\star + \sum_{k=1}^n \gamma \rho \zeta (2 - \tfrac{1}{2} L_1 \gamma \rho) \sqrt{\mathbb{E}[\|\lambda^k\|_2^2]} + \gamma \tfrac{1}{2} \zeta (1 - L_1 \gamma \rho) \mathbb{E}[\|\lambda^k\|_2^2]$$

where the second to last inequality is due to Jensen's inequality and the last inequality uses that $\gamma \leq 1/\rho L_1$. Using the stronger requirement that $\gamma \leq 1/2\rho L_1$ it follows that

$$\tfrac{1}{4} \gamma \left( \rho \sum_{k \in \mathcal{A}} \mathbb{E}[\|\nabla f(x^k)\|_*] + \sum_{k \notin \mathcal{A}} \mathbb{E}[\|\nabla f(x^k)\|_*^2] \right) \leq \Delta + \gamma \epsilon_n$$

with $\Delta := f(x^1) - f^\star$ and $\epsilon_n := \tfrac{1}{n} \sum_{k=1}^n \rho \zeta \tfrac{7}{4} \sqrt{\mathbb{E}[\|\lambda^k\|_2^2]} + \tfrac{1}{4} \zeta \mathbb{E}[\|\lambda^k\|_2^2]$. By nonnegativity of the summands, it follows that

$$\tfrac{1}{n} \sum_{k \in \mathcal{A}} \mathbb{E}[\|\nabla f(x^k)\|_*] \leq \tfrac{4\Delta}{\gamma \rho n} + \tfrac{4\epsilon_n}{\rho}, \tag{15}$$

corresponding to the indices from Case I. Using Jensen's inequality, we similarly have

$$\tfrac{1}{n} \sum_{k \notin \mathcal{A}} \mathbb{E}[\|\nabla f(x^k)\|_*]^2 \leq \tfrac{1}{n} \sum_{k \notin \mathcal{A}} \mathbb{E}[\|\nabla f(x^k)\|_*^2] \leq \tfrac{4\Delta}{\gamma n} + 4\epsilon_n =: A \tag{16}$$

corresponding to the indices from Case II. We will now use that $2az - a^2 \leq z^2$ for any $a, z > 0$. Pick $z = \mathbb{E}[\|\nabla f(x^k)\|_*]$, then we have that

$$\tfrac{1}{n} \sum_{k \notin \mathcal{A}} z \leq \tfrac{1}{n} \sum_{k \notin \mathcal{A}} \tfrac{z^2}{2a} + \tfrac{a}{2} \overset{(16)}{\leq} \tfrac{A}{2a} + \tfrac{1}{n} \sum_{k \notin \mathcal{A}} \tfrac{a}{2} \leq \tfrac{A}{2a} + \tfrac{a}{2}$$

Choosing $a = \sqrt{A}$ and using the triangle inequality we have

$$\frac{1}{n}\sum_{k\notin\mathcal{A}}\mathbb{E}[\|\nabla f(x^k)\|_*] \leq \sqrt{A} \leq \frac{2\sqrt{\Delta}}{\sqrt{n}} + 2\sqrt{\epsilon_n}. \tag{17}$$

Summing the two cases, (15) and (17), we have

$$\frac{1}{n}\sum_{k=1}^{n}\mathbb{E}[\|\nabla f(x^k)\|_*] \leq \frac{4\sqrt{\Delta}}{\sqrt{\gamma n}} + \frac{8\Delta}{\gamma\rho n} + 4\sqrt{\epsilon_n} + \frac{8\epsilon_n}{\rho}.$$

What remains is to bound the error $\epsilon_n$ that is due to the stochastic estimator. With the choice $\gamma \leq 1/\sqrt{n}L_0$, $\gamma\rho \leq 1/2n^{3/4}L_1$ and $\alpha_k = 1/\sqrt{n}$, invoke Lemma B.2 from which we have

$$\mathbb{E}[\|\lambda^k\|_2^2] \leq \frac{2(\sigma^2 + \zeta_*^2 L^2/L_1^2)}{\sqrt{n}} =: B.$$

It follows that

$$\epsilon_n \leq \rho\zeta\frac{7}{4}\sqrt{B} + \frac{1}{4}\zeta B$$

and in turn the inequality B.2.1 simplifies

$$\frac{1}{n}\sum_{k=1}^{n}\mathbb{E}[\|\nabla f(x^k)\|_*] \leq O\left(\frac{\sqrt{\Delta}}{\sqrt{\gamma n}} + \frac{\Delta}{\gamma\rho n} + \sqrt{\rho\zeta\sqrt{B} + \zeta B} + \frac{\rho\zeta\sqrt{B} + \zeta B}{\rho}\right)$$

$$\leq O\left(\frac{\sqrt{\Delta}}{\sqrt{\gamma n}} + \frac{\Delta}{\gamma\rho n} + \sqrt{\rho\zeta}B^{1/4} + \sqrt{\zeta B} + \zeta\sqrt{B} + \frac{\zeta B}{\rho}\right)$$

where we have used the triangle inequality in the second inequality. Letting $b := 2(\sigma^2 + \zeta_*^2 L^2/L_1^2)$ we have with the choice $\gamma = 1/\sqrt{n}L_0$ and $\rho = L_0/2n^{1/4}L_1$ that

$$\frac{1}{n}\sum_{k=1}^{n}\mathbb{E}[\|\nabla f(x^k)\|_*] \leq O\left(\frac{1}{n^{1/4}}\left(\sqrt{\Delta L_0} + \Delta L_1 + \frac{\sqrt{\zeta L_0}b^{1/4}}{\sqrt{L_1}} + \sqrt{\zeta b} + \zeta\sqrt{b} + \frac{\zeta b L_1}{L_0}\right)\right)$$

$$\leq O\left(\frac{1}{n^{1/4}}\left(\sqrt{\Delta L_0} + \Delta L_1 + \frac{\sqrt{\zeta L_0}(\sigma^2 + \zeta_*^2 L^2/L_1^2)^{1/4}}{\sqrt{L_1}}\right.\right.$$

$$\left.\left. + (\zeta + \sqrt{\zeta})\sqrt{(\sigma^2 + \zeta_*^2 L^2/L_1^2)} + \frac{\zeta(\sigma^2 + \zeta_*^2 L^2/L_1^2)L_1}{L_0}\right)\right)$$

$$\leq O\left(\frac{1}{n^{1/4}}\left(\sqrt{\Delta L_0} + \Delta L_1 + \frac{\sqrt{\zeta L_0}(\sqrt{\sigma} + \sqrt{\zeta_* L/L_1})}{\sqrt{L_1}}\right.\right.$$

$$\left.\left. + (\zeta + \sqrt{\zeta})(\sigma + \zeta_* L/L_1) + \frac{\zeta(\sigma^2 + \zeta_*^2 L^2/L_1^2)L_1}{L_0}\right)\right)$$

Noting that $\mathbb{E}[\|\nabla f(\bar{x}^n)\|_*] = \frac{1}{n}\sum_{k=1}^{n}\mathbb{E}[\|\nabla f(x^k)\|_*]$ completes the proof. $\square$

### B.2.2  Convergence analysis of S$^3$CG

Following the same outline as the convergence analysis for Algorithm 1 given in the previous subsection, we start with an error control lemma in the vein of [Mokhtari et al., 2020, Lem. 6] that is compatible with our adaptive stepsize.

**Lemma B.3** (Linear recursive inequality for $\mathbb{E}\|\lambda^k\|_2^2$ for S$^3$CG). *Suppose Assumptions 4.2 and 4.8 hold and let $n \in \mathbb{N}^*$. Consider the iterates $\{x^k\}_{1\leq k\leq n}$ generated by Algorithm 2 with stepsize $\gamma\eta_k \leq \rho$. Then, for all $k \in \{1,\ldots,n\}$,*

$$\mathbb{E}[\|\lambda^k\|_2^2] \leq \left(1 - \frac{\alpha}{2}\right)\mathbb{E}[\|\lambda^{k-1}\|_2^2] + \alpha^2\sigma^2 + \frac{8\zeta_*^2 L^2\beta\gamma^2\rho^2}{\alpha}$$

*where $\zeta_* := \max_{x\in\mathcal{X}}\frac{\|x\|_2}{\|x\|_*}$.*

*Proof.* The proof is a straightforward adaptation of the arguments laid out in Mokhtari et al. [2020, Lem. 6], which in fact do not depend on convexity of the function $f$ nor on the choice of stepsize

$\gamma\eta_k$, as long as it is in $[0, 1]$. Let $n \in \mathbb{N}^*$ and $k \in \{1, \ldots, n\}$, then

$$
\begin{aligned}
\left\|\lambda^k\right\|_2^2 &= \left\|\nabla f(x^k) - d^k\right\|_2^2 \\
&= \left\|\nabla f(x^k) - \alpha\nabla f(x^k, \xi_k) - (1-\alpha)d^{k-1}\right\|_2^2 \\
&= \left\|\alpha\left(\nabla f(x^k) - \nabla f(x^k, \xi_k)\right) + (1-\alpha)\left(\nabla f(x^k) - \nabla f(x^{k-1})\right) - (1-\alpha)\left(d^{k-1} - \nabla f(x^{k-1})\right)\right\|_2^2 \\
&= \alpha^2\left\|\nabla f(x^k) - \nabla f(x^k, \xi_k)\right\|_2^2 + (1-\alpha)^2\left\|\nabla f(x^k) - \nabla f(x^{k-1})\right\|_2^2 \\
&\quad + (1-\alpha)^2\left\|\nabla f(x^{k-1}) - d^{k-1}\right\|_2^2 \\
&\quad + 2\alpha(1-\alpha)\langle\nabla f(x^{k-1}) - \nabla f(x^{k-1}, \xi_{k-1}), \nabla f(x^k) - \nabla f(x^{k-1})\rangle \\
&\quad + 2\alpha(1-\alpha)\langle\nabla f(x^k) - \nabla f(x^k, \xi_k), \nabla f(x^{k-1}) - d^{k-1}\rangle \\
&\quad + 2(1-\alpha)^2\langle\nabla f(x^k) - \nabla f(x^{k-1}), \nabla f(x^{k-1}) - d^{k-1}\rangle.
\end{aligned}
$$

Taking the expectation conditioned on the filtration $\mathcal{F}_k$ generated by the iterates until $k$, i.e., the sigma algebra generated by $\{x^1, \ldots, x^k\}$, which we denote using $\mathbb{E}_k[\cdot]$, and using the unbiased property in Assumption 4.8, we get,

$$
\begin{aligned}
\mathbb{E}_k[\left\|\lambda^k\right\|_2^2] &= \alpha^2\mathbb{E}_k[\left\|\nabla f(x^k) - \nabla f(x^k, \xi_k)\right\|_2^2] + (1-\alpha)^2\left\|\nabla f(x^k) - \nabla f(x^{k-1})\right\|_2^2 \\
&\quad + (1-\alpha)^2\left\|\lambda^{k-1}\right\|_2^2 + 2(1-\alpha)^2\langle\nabla f(x^k) - \nabla f(x^{k-1}), \lambda^{k-1}\rangle.
\end{aligned}
$$

From the above expression we can estimate,

$$
\begin{aligned}
\mathbb{E}_k[\left\|\lambda^k\right\|_2^2] &\overset{(a)}{\leq} \alpha^2\sigma^2 + (1-\alpha)^2\left\|\nabla f(x^k) - \nabla f(x^{k-1})\right\|_2^2 + (1-\alpha)^2\left\|\lambda^{k-1}\right\|_2^2 \\
&\quad + 2(1-\alpha)^2\langle\nabla f(x^k) - \nabla f(x^{k-1}), \lambda^{k-1}\rangle \\
&\overset{(b)}{\leq} \alpha^2\sigma^2 + (1-\alpha)^2\left\|\nabla f(x^k) - \nabla f(x^{k-1})\right\|_2^2 + (1-\alpha)^2\left\|\lambda^{k-1}\right\|_2^2 \\
&\quad + (1-\alpha)^2\left(\tfrac{\alpha}{2}\left\|\nabla f(x^k) - \nabla f(x^{k-1})\right\|_2^2 + \tfrac{2}{\alpha}\left\|\lambda^{k-1}\right\|_2^2\right) \\
&\overset{(c)}{\leq} \alpha^2\sigma^2 + (1-\alpha)^2\zeta_*^2\left\|\nabla f(x^k) - \nabla f(x^{k-1})\right\|^2 + (1-\alpha)^2\left\|\lambda^{k-1}\right\|_2^2 \\
&\quad + (1-\alpha)^2\left(\tfrac{\alpha}{2}\zeta_*^2\left\|\nabla f(x^k) - \nabla f(x^{k-1})\right\|^2 + \tfrac{2}{\alpha}\left\|\lambda^{k-1}\right\|_2^2\right) \\
&\overset{(d)}{\leq} \alpha^2\sigma^2 + (1-\alpha)^2\zeta_*^2 L^2\left\|x^k - x^{k-1}\right\|^2 + (1-\alpha)^2\left\|\lambda^{k-1}\right\|_2^2 \\
&\quad + (1-\alpha)^2\left((\tfrac{\alpha}{2})\zeta_* L^2\left\|x^k - x^{k-1}\right\|^2 + \tfrac{2}{\alpha}\left\|\lambda^{k-1}\right\|_2^2\right) \\
&\overset{(e)}{\leq} \alpha^2\sigma^2 + 4(1-\alpha)^2\zeta_*^2 L^2\beta^2\gamma^2\eta_k^2 + (1-\alpha)^2\left\|\lambda^{k-1}\right\|_2^2 + (1-\alpha)^2\left(2\alpha\zeta_*^2 L^2\beta^2\gamma^2\eta_k^2 + \tfrac{2}{\alpha}\left\|\lambda^{k-1}\right\|_2^2\right) \\
&\overset{(f)}{\leq} \alpha^2\sigma^2 + 4(1+\tfrac{\alpha}{2})(1-\alpha)\zeta_*^2 L^2\beta^2\gamma^2\eta_k^2 + (1+\tfrac{2}{\alpha})(1-\alpha)\left\|\lambda^{k-1}\right\|_2^2 \\
&\overset{(g)}{\leq} \alpha^2\sigma^2 + 4(1+\tfrac{\alpha}{2})(1-\alpha)\zeta_*^2 L^2\beta^2\gamma^2\rho^2 + (1+\tfrac{2}{\alpha})(1-\alpha)\left\|\lambda^{k-1}\right\|_2^2,
\end{aligned}
$$

using the bounded variance property from Assumption 4.8 for (a), Young's inequality with parameter $\alpha/2 > 0$ for (b), $\zeta_* := \max_{x \in \mathcal{X}} \frac{\|x\|_2}{\|x\|_*}$ for (c), Assumption 4.2 for (d), the definition of $x^k$ from Algorithm 2 for (e), the fact that $1 - \alpha < 1$ for (f), and $\eta_k \leq \rho$ and for (g). To complete the proof, we note that

$$
(1 + \tfrac{2}{\alpha})(1 - \alpha) \leq (1 - \tfrac{\alpha}{2}) \quad \text{and} \quad (1 - \alpha)(1 + \tfrac{\alpha}{2}) \leq \tfrac{2}{\alpha}
$$

which, applied to the previous inequality and taking total expectations, yields

$$
\mathbb{E}[\left\|\lambda^k\right\|_2^2] \leq \left(1 - \frac{\alpha}{2}\right)\mathbb{E}[\left\|\lambda^{k-1}\right\|_2^2] + \alpha^2\sigma^2 + \frac{8\zeta_*^2 L^2\beta^2\gamma^2\rho^2}{\alpha}.
$$

$\square$

**Lemma B.4** (Bound on the gradient error with horizon-dependent $\alpha$ for S³CG). *Suppose Assumption 4.8 holds, $f$ is $L$-smooth with respect to $\|\cdot\|_*$, and let $n \in \mathbb{N}^*$. Consider the iterates $\{x^k\}_{1 \le k \le n}$ generated by Algorithm 2 with a stepsize $\gamma\rho$ satisfying*

$$\gamma\rho < \tfrac{1}{Ln^{3/4}}. \tag{18}$$

*Moreover, consider a constant momentum $\alpha_k = \alpha = \frac{1}{\sqrt{n}}$ for all $k \in \{1, \ldots, n\}$. Then, for all $k \in \{1, \ldots, n\}$ the following holds*

$$\mathbb{E}[\|\lambda^k\|_2^2] \le \tfrac{2\sigma^2 + 16\zeta_*^2\beta^2}{\sqrt{n}}. \tag{19}$$

*Proof.* Let $n \in \mathbb{N}^*$ and $k \in \{1, \ldots, n\}$. We start from the recursive inequality obtained in Lemma B.3 for $\mathbb{E}[\|\lambda^k\|_2^2]$ with $L$ the Lipschitz constant of the gradient over the compact set $\mathcal{D}$ to get

$$\mathbb{E}[\|\lambda^k\|_2^2] \le \left(1 - \tfrac{\alpha}{2}\right)\mathbb{E}[\|\lambda^{k-1}\|_2^2] + \alpha^2\sigma^2 + \frac{8\zeta_*^2 L^2\beta^2\gamma^2\rho^2}{\alpha}. \tag{20}$$

Now, we substitute the specific choice $\alpha = \frac{1}{\sqrt{n}}$:

$$\mathbb{E}[\|\lambda^k\|_2^2] \le \left(1 - \frac{1}{2\sqrt{n}}\right)\mathbb{E}[\|\lambda^{k-1}\|_2^2] + \frac{\sigma^2}{n} + 8\zeta_*^2 L^2\beta^2\gamma^2\rho^2\sqrt{n}. \tag{21}$$

Using the particular choice of $\gamma\rho$ specified in the statement of the lemma, we have

$$\mathbb{E}[\|\lambda^k\|_2^2] \le \left(1 - \frac{1}{2\sqrt{n}}\right)\mathbb{E}[\|\lambda^{k-1}\|_2^2] + \frac{1}{n}(\sigma^2 + 8\zeta_*^2\beta^2)$$

Let $u_k = \mathbb{E}[\|\lambda^k\|_2^2]$, $a = \frac{1}{2\sqrt{n}}$, and $b = \frac{1}{n}(\sigma^2 + 8\zeta_*^2\beta^2)$. Unrolling the recurrence relation $u_k \le (1-a)u_{k-1} + b$, we have

$$u_k \le (1-a)^k u_0 + b\sum_{i=0}^{k-1}(1-a)^i = b\sum_{i=0}^{k-1}(1-a)^i = b\frac{1-(1-a)^k}{1-(1-a)} = b\frac{1-(1-a)^k}{a}$$

Since $0 < a < 1$, we have $0 < (1-a)^k < 1$ for $k \ge 1$. Thus, $1 - (1-a)^k < 1$. Therefore,

$$u_k \le \tfrac{b}{a}. \tag{22}$$

Substituting the values for $a$ and $b$, we have

$$\mathbb{E}[\|\lambda^k\|_2^2] \le \tfrac{2\sigma^2 + 16\zeta_*^2\beta^2}{\sqrt{n}}. \tag{23}$$

This concludes the proof. $\qquad\square$

**Theorem 4.11.** *Suppose Assumptions 4.2 and 4.8 hold and let $n \in \mathbb{N}^*$. Consider the iterates $\{x^k\}_{1 \le k \le n}$ generated by Algorithm 2 (Variant 1) with a constant stepsize $\gamma \le 1/L\sqrt{n}$ and $\rho \le 1/n^{1/4}$. Then, for all $u \in \beta\mathcal{D}$,*

$$\mathbb{E}[\langle \nabla f(\bar{x}^n), \bar{x}^n - u\rangle] \le \tfrac{4\sqrt{\Delta}}{\sqrt{\gamma n}} + \tfrac{8\Delta}{\gamma\rho n} + 4\sqrt{\epsilon_n} + \tfrac{8\epsilon_n}{\rho}$$

*where $\Delta := f(x^1) - f^\star$ and $\epsilon_n := \frac{1}{n}\sum_{k=1}^n O(\sqrt{\mathbb{E}\|\lambda^k\|_2^2} + \mathbb{E}\|\lambda^k\|_2^2)$.*

*Furthermore, taking $\alpha = 1/\sqrt{n}$, $\gamma = 1/(L\sqrt{n})$ and $\rho = 1/n^{1/4}$ such that $\gamma\rho = 1/(Ln^{3/4})$ we have that*

$$\mathbb{E}[\langle \nabla f(\bar{x}^n), \bar{x}^n - u\rangle] \le O\left(\tfrac{1}{n^{1/4}}\right).$$

*Proof.* Note that, since $f$ is continuously differentiable and $\mathcal{D}$ is compact, $f$ must be Lipschitz-smooth on the scaled ball $\beta\mathcal{D}$ with respect to the norm $\|\cdot\|$; call the Lipschitz constant $L > 0$. We can therefore start with the descent lemma for $f$ at the points $x^{k+1}$ and $x^k$ to find

$$0 \le f(x^k) - f(x^{k+1}) + \langle \nabla f(x^k), x^{k+1} - x^k\rangle + \frac{L}{2}\|x^{k+1} - x^k\|^2$$

$$\le f(x^k) - f(x^{k+1}) - \gamma\eta_k\langle \nabla f(x^k), v^k\rangle + \frac{L}{2}\gamma^2\eta_k^2\|v^k\|^2$$

$$\le f(x^k) - f(x^{k+1}) - \gamma\eta_k\left(\langle d^k, v^k\rangle + \langle \nabla f(x^k) - d^k, v^k\rangle\right) + \frac{L}{2}\gamma^2\eta_k^2\|v^k\|^2.$$

Now we can proceed case-by-case depending on whether clipping is active or not.

**Case I**  Clipping Active $(\gamma\eta_k = \gamma\rho;\ \frac{\langle d^k, v^k\rangle}{\|v^k\|^2} \geq \rho)$.

For all $u \in \beta\mathcal{D}$ it holds,

$$0 \leq f(x^k) - f(x^{k+1}) - \gamma\eta_k \left(\langle d^k, v^k\rangle + \langle \nabla f(x^k) - d^k, v^k\rangle\right) + \frac{L}{2}\gamma^2\eta_k^2\|v^k\|^2$$

$$\leq f(x^k) - f(x^{k+1}) - \gamma\eta_k\left(\frac{1}{2}\langle d^k, v^k\rangle + \frac{1}{2}\langle\nabla f(x^k), x^k - u\rangle + \frac{1}{2}\langle d^k - \nabla f(x^k), x^k - u\rangle + \langle\nabla f(x^k) - d^k, v^k\rangle\right) + \frac{L}{2}\gamma^2\eta_k^2\|v^k\|^2$$

$$\overset{(a)}{\leq} f(x^k) - f(x^{k+1}) - \gamma\eta_k\left(\frac{1}{2}\langle d^k, v^k\rangle + \frac{1}{2}\langle\nabla f(x^k), x^k - u\rangle - \frac{1}{2}\|\lambda^k\|_2\|x^k - u\|_2 - \|\lambda^k\|_2\|v^k\|_2\right) + \frac{L}{2}\gamma^2\eta_k^2\|v^k\|^2$$

$$\overset{(b)}{\leq} f(x^k) - f(x^{k+1}) + \gamma\eta_k\left(\frac{L}{2}\gamma\eta_k\|v^k\|^2 - \frac{1}{2}\langle d^k, v^k\rangle\right) - \gamma\eta_k\frac{1}{2}\langle\nabla f(x^k), x^k - u\rangle + \gamma\eta_k\frac{3}{2}\|\lambda^k\|_2 D_2$$

$$\overset{(c)}{\leq} f(x^k) - f(x^{k+1}) + \gamma\rho\left(\frac{L}{2}\gamma\rho\|v^k\|^2 - \frac{1}{2}\rho\|v^k\|^2\right) - \gamma\rho\frac{1}{2}\langle\nabla f(x^k), x^k - u\rangle + \gamma\rho\frac{3}{2}\|\lambda^k\|_2 D_2$$

$$\overset{(d)}{\leq} f(x^k) - f(x^{k+1}) + \gamma\rho\frac{3}{2}\|\lambda^k\|_2 D_2 - \gamma\rho\frac{1}{2}\langle\nabla f(x^k), x^k - u\rangle$$

where $D_2 = \max\limits_{x,y\in\beta\mathcal{D}}\|x - y\|_2$ is the diameter of the set $\beta\mathcal{D}$ in the Euclidean norm. The inequality (a) follows by Cauchy-Schwarz, (b) follows by using the diameter of $\beta\mathcal{D}$, (c) follows since clipping is active, and (d) follows since $L\gamma \leq 1$. Finally, rearranging gives

$$\gamma\rho\langle\nabla f(x^k), x^k - u\rangle \leq 2\left(f(x^k) - f(x^{k+1})\right) + 3D_2\gamma\rho\|\lambda^k\|_2. \tag{24}$$

**Case II**  No Clipping $(\gamma\eta_k = \gamma\frac{\langle d^k, v^k\rangle}{\|v^k\|^2};\ \frac{\langle d^k, v^k\rangle}{\|v^k\|^2} \leq \rho)$.

In this case, our stepsize acts like the short step. Starting with the previous inequality from the descent lemma we have, for all $u \in \beta\mathcal{D}$,

$$0 \leq f(x^k) - f(x^{k+1}) - \gamma\frac{\langle d^k, v^k\rangle}{\|v^k\|^2}\langle d^k, v^k\rangle - \gamma\eta_k\langle\nabla f(x^k) - d^k, v^k\rangle + \frac{L}{2}\gamma^2\left(\frac{\langle d^k, v^k\rangle}{\|v^k\|^2}\right)^2\|v^k\|^2$$

$$\leq f(x^k) - f(x^{k+1}) - \gamma\frac{\langle d^k, v^k\rangle^2}{\|v^k\|^2} - \gamma\eta_k\langle\nabla f(x^k) - d^k, v^k\rangle + L\gamma^2\frac{\langle d^k, v^k\rangle^2}{2\|v^k\|^2} \tag{25}$$

$$\leq f(x^k) - f(x^{k+1}) - \gamma\frac{\langle d^k, v^k\rangle^2}{2\|v^k\|^2} - \gamma\eta_k\langle\nabla f(x^k) - d^k, v^k\rangle$$

where in the last inequality we have used that $\gamma \leq \frac{1}{L}$. Rearranging, we can estimate

$$0 \leq f(x^k) - f(x^{k+1}) - \gamma\frac{\langle d^k, v^k\rangle^2}{2\|v^k\|^2} - \gamma\eta_k\langle\nabla f(x^k) - d^k, v^k\rangle$$

$$\overset{(a)}{\leq} f(x^k) - f(x^{k+1}) - \frac{1}{2\|v^k\|^2}\left(\frac{\gamma}{2}\langle\nabla f(x^k), x^k - u\rangle^2 - 2\gamma\langle\nabla f(x^k) - d^k, x^k - u\rangle^2\right) - \gamma\eta_k\langle\nabla f(x^k) - d^k, v^k\rangle$$

$$= f(x^k) - f(x^{k+1}) - \frac{\gamma}{4\|v^k\|^2}\langle\nabla f(x^k), x^k - u\rangle^2 + \frac{\gamma}{\|v^k\|^2}\langle\nabla f(x^k) - d^k, x^k - u\rangle^2 - \gamma\eta_k\langle\nabla f(x^k) - d^k, v^k\rangle$$

$$\overset{(b)}{\leq} f(x^k) - f(x^{k+1}) - \frac{\gamma}{4\|v^k\|^2}\langle\nabla f(x^k), x^k - u\rangle^2 + \frac{D_2^2\gamma}{\|v^k\|^2}\|\lambda^k\|_2^2 + D_2\gamma\rho\|\lambda^k\|_2$$

$$\overset{(c)}{\leq} 16\beta^2\left(f(x^k) - f(x^{k+1})\right) - \gamma\langle\nabla f(x^k), x^k - u\rangle^2 + 4D_2^2\gamma\|\lambda^k\|_2^2 + 16D_2\beta^2\gamma\rho\|\lambda^k\|_2$$

$$\tag{26}$$

where (a) is due to Young's inequality, (b) is due to Cauchy-Schwarz and the definition of $D_2$ as the diameter of $\beta\mathcal{D}$ in the Euclidean norm, and (c) follows by multiplying everything by $4\|v^k\|^2$ and estimating. We can rearrange this to finally arrive at

$$\gamma\langle\nabla f(x^k), x^k - u\rangle^2 \leq 16\beta^2\left(f(x^k) - f(x^{k+1})\right) + 4D_2^2\gamma\|\lambda^k\|_2^2 + 16D_2\beta^2\gamma\rho\|\lambda^k\|_2.$$

**Combining Both Cases**  Let $M = \max\{16\beta^2, 2\}$ and $M' = \max\{16D_2\beta^2, 3D_2\}$ and let $\mathcal{A} \subset \{1, 2, \ldots, n\}$ denote the indices where clipping is active. Let $n \in \mathbb{N}^*$ and denote

$$\epsilon_n := \frac{1}{n}\sum_{k=1}^{n} M'\rho\mathbb{E}[\|\lambda^k\|_2] + \frac{1}{n}\sum_{k=1}^{n} 4D_2^2\mathbb{E}[\|\lambda^k\|_2^2].$$

Then, taking expectations, adding from $k = 1$ to $n$, and dividing by $n$ gives

$$\frac{1}{n}\sum_{k\in\mathcal{A}}\gamma\rho\mathbb{E}[\langle\nabla f(x^k), x^k - u\rangle] + \frac{1}{n}\sum_{k\notin\mathcal{A}}\gamma\mathbb{E}[\langle\nabla f(x^k), x^k - u\rangle^2] \le \frac{M}{n}\left(f(x^1) - f^\star\right) + \gamma\epsilon_n. \tag{27}$$

We can lower bound the left hand side by the sum over $\mathcal{A}$ and divide by $\gamma\rho$ to get

$$\frac{1}{n}\sum_{k\in\mathcal{A}}\mathbb{E}[\langle\nabla f(x^k), x^k - u\rangle] \le \frac{M\Delta}{\gamma\rho n} + \frac{\epsilon_n}{\rho}. \tag{28}$$

Similarly, lower bounding the left hand side by the sum over the complement of $\mathcal{A}$ and dividing by $\gamma$, we get by Jensen's inequality

$$\frac{1}{n}\sum_{k\notin\mathcal{A}}\mathbb{E}[\langle\nabla f(x^k), x^k - u\rangle]^2 \le \frac{1}{n}\sum_{k\notin\mathcal{A}}\mathbb{E}[\langle\nabla f(x^k), x^k - u\rangle^2] \le \frac{M\Delta}{\gamma n} + \epsilon_n. \tag{29}$$

We use the fact that $2az - a^2 \le z^2$ for any $a, z > 0$. Picking $z = \mathbb{E}[\langle\nabla f(x^k), x^k - u\rangle]$, it follows that

$$\frac{1}{n}\sum_{k\notin\mathcal{A}}z \le \frac{1}{n}\sum_{k\notin\mathcal{A}}\frac{z^2}{2a} + \frac{a}{2} \le \frac{A}{2a} + \frac{1}{n}\sum_{k\notin\mathcal{A}}\frac{a}{2} \le \frac{A}{2a} + \frac{a}{2} \tag{30}$$

where $A := \frac{M\Delta}{\gamma n} + \epsilon_n$. Choosing $a = \sqrt{A}$ and replacing $z$ by $\mathbb{E}[\langle\nabla f(x^k), x^k - u\rangle]$ we get

$$\frac{1}{n}\sum_{k\notin\mathcal{A}}\mathbb{E}[\langle\nabla f(x^k), x^k - u\rangle] \le \sqrt{A} = \frac{\sqrt{M\Delta}}{\sqrt{\gamma n}} + \sqrt{\epsilon_n}. \tag{31}$$

Adding both of these we get

$$\frac{1}{n}\sum_{k=1}^{n}\mathbb{E}[\langle\nabla f(x^k), x^k - u\rangle] \le \frac{\sqrt{M\Delta}}{\sqrt{\gamma n}} + \sqrt{\epsilon_n} + \frac{M\Delta}{\gamma\rho n} + \frac{\epsilon_n}{\rho}. \tag{32}$$

Let $\Lambda_n^2 := \frac{2\sigma^2 + 16\zeta_*^2\beta^2}{\sqrt{n}}$. By lemma B.4, $\Lambda_n^2 \ge \mathbb{E}[\|\lambda^k\|_2^2]$ for all $k \le n$.

Next, we can estimate

$$\epsilon_n \le M'\rho\Lambda_n + 4D_2^2\Lambda_n^2$$
$$\sqrt{\epsilon_n} \le \sqrt{M'\rho\Lambda_n} + 2D_2\Lambda_n$$
$$\frac{\epsilon_n}{\rho} \le M'\Lambda_n + \frac{4}{\rho}D_2^2\Lambda_n^2.$$

Substituting in the definition of $\Lambda_n$, $\gamma$, and $\rho$ while also noting the definition of $\bar{x}^n$, we get

$$\mathbb{E}[\langle\nabla f(\bar{x}^n), \bar{x}^n - u\rangle] \le \frac{\sqrt{M\Delta}}{\sqrt{\gamma n}} + \sqrt{\epsilon_n} + \frac{M\Delta}{\gamma\rho n} + \frac{\epsilon_n}{\rho}$$

$$\le \frac{\sqrt{M\Delta}}{\sqrt{\gamma n}} + \sqrt{M'\rho\Lambda_n} + 2D_2\Lambda_n + \frac{M\Delta}{\gamma\rho n} + M'\Lambda_n + \frac{4}{\rho}D_2^2\Lambda_n^2$$

$$\le \frac{\sqrt{LM\Delta}}{n^{1/4}} + \frac{\sqrt{M'\left(2\sigma^2 + 16\zeta_*^2\beta^2\right)}}{n^{1/4}} + \frac{2D_2\sqrt{2\sigma^2 + 16\zeta_*^2\beta^2}}{n^{1/4}}$$

$$+ \frac{LM\Delta}{n^{1/4}} + \frac{M'\sqrt{2\sigma^2 + 16\zeta_*^2\beta^2}}{n^{1/4}} + \frac{4D_2^2\left(2\sigma^2 + 16\zeta_*^2\beta^2\right)}{n^{1/4}}$$

which gives a big O rate

$$\mathbb{E}[\langle\nabla f(\bar{x}^n, \bar{x}^n - u\rangle] \le O\left(\frac{1}{n^{1/4}}\right).$$

$\square$

**Theorem B.5.** *Suppose Assumptions 4.2 and 4.8 hold and let $n \in \mathbb{N}^*$. Consider the iterates $\{x^k\}_{1 \le k \le n}$ generated by Algorithm 2 (Variant 2) with a constant stepsize $\gamma \le {}^1/_{L\sqrt{n}}$ and $\rho \le {}^1/_{n^{1/4}}$. Then, for all $u \in \mathcal{D}$,*

$$\mathbb{E}[\langle \nabla f(\bar{x}^n), \bar{x}^n - u \rangle] \le \frac{4\sqrt{\Delta}}{\sqrt{\gamma n}} + \frac{8\Delta}{\gamma\rho n} + 4\sqrt{\epsilon_n} + \frac{8\epsilon_n}{\rho}$$

*where $\Delta := f(x^1) - f^\star$ and $\epsilon_n := \frac{1}{n}\sum_{k=1}^n O(\sqrt{\mathbb{E}\|\lambda^k\|_2^2} + \mathbb{E}\|\lambda^k\|_2^2)$.*

*Furthermore, taking $\alpha = {}^1/\sqrt{n}$, $\gamma = {}^1/(L\sqrt{n})$ and $\rho = {}^1/n^{1/4}$ such that $\gamma\rho = {}^1/(Ln^{3/4})$ we have that*

$$\mathbb{E}[\langle \nabla f(\bar{x}^n), \bar{x}^n - u \rangle] \le O(\tfrac{1}{n^{1/4}}).$$

*Proof.* Note that, since $f$ is continuously differentiable and $\mathcal{D}$ is compact, $f$ must be Lipschitz-smooth on the scaled ball $\beta\mathcal{D}$ with respect to the norm $\|\cdot\|$; call the Lipschitz constant $L$. We can therefore start with the descent lemma for $f$ at the points $x^{k+1}$ and $x^k$ to find

$$0 \le f(x^k) - f(x^{k+1}) + \langle \nabla f(x^k), x^{k+1} - x^k \rangle + \frac{L}{2}\|x^{k+1} - x^k\|^2$$

$$\le f(x^k) - f(x^{k+1}) - \gamma\eta_k\langle \nabla f(x^k), v^k \rangle + \frac{L}{2}\gamma^2\eta_k^2\|v^k\|^2 \qquad (33)$$

$$\le f(x^k) - f(x^{k+1}) - \gamma\eta_k\left(\langle d^k, v^k \rangle + \langle \nabla f(x^k) - d^k, v^k \rangle\right) + \frac{L}{2}\gamma^2\eta_k^2\|v^k\|^2.$$

Now we can proceed case-by-case depending on whether clipping is active or not.

**Case I** Clipping Active $(\gamma\eta_k = \gamma\rho; \frac{\langle d^k, v^k \rangle}{4\beta^2} \ge \rho)$.

$$0 \le f(x^k) - f(x^{k+1}) - \gamma\eta_k\left(\langle d^k, v^k \rangle + \langle \nabla f(x^k) - d^k, v^k \rangle\right) + \frac{L}{2}\gamma^2\eta_k^2\|v^k\|^2$$

$$\le f(x^k) - f(x^{k+1}) - \gamma\eta_k\left(\frac{1}{2}\langle d^k, v^k \rangle + \frac{1}{2}\langle \nabla f(x^k), x^k - u \rangle + \frac{1}{2}\langle d^k - \nabla f(x^k), x^k - u \rangle + \langle \nabla f(x^k) - d^k, v^k \rangle\right) + \frac{L}{2}\gamma^2\eta_k^2\|v^k\|^2$$

$$\le f(x^k) - f(x^{k+1}) - \gamma\eta_k\left(\frac{1}{2}\langle d^k, v^k \rangle + \frac{1}{2}\langle \nabla f(x^k), x^k - u \rangle - \frac{1}{2}\|\lambda^k\|_2\|x^k - u\|_2 - \|\lambda^k\|_2\|v^k\|_2\right) + \frac{L}{2}\gamma^2\eta_k^2\|v^k\|^2$$

$$\le f(x^k) - f(x^{k+1}) + \gamma\eta_k\left(\frac{L}{2}\gamma\eta_k\|v^k\|^2 - \frac{1}{2}\langle d^k, v^k \rangle\right) - \gamma\eta_k\frac{1}{2}\langle \nabla f(x^k), x^k - u \rangle + \gamma\eta_k\frac{3}{2}\|\lambda^k\|_2 D_2$$

$$\le f(x^k) - f(x^{k+1}) + \gamma\rho\left(2L\gamma\rho\beta^2 - 2\rho\beta^2\right) - \gamma\rho\frac{1}{2}\langle \nabla f(x^k), x^k - u \rangle + \gamma\rho\frac{3}{2}\|\lambda^k\|_2 D_2$$

$$\le f(x^k) - f(x^{k+1}) + \gamma\rho\frac{3}{2}\|\lambda^k\|_2 D_2 - \gamma\rho\frac{1}{2}\langle \nabla f(x^k), x^k - u \rangle$$

$$(34)$$

in the second inequality we have used the fact that $\|v^k\|^2 \le 2\|x^k\|^2 + 2\|\beta\operatorname{Imo}(d^k)\|^2 \le 4\beta^2$ and the fact that $\|v^k\|_2 \le D_2$. Finally, rearranging gives

$$\gamma\rho\langle \nabla f(x^k), x^k - u \rangle \le 2\left(f(x^k) - f(x^{k+1})\right) + 3D_2\gamma\rho\|\lambda^k\|_2. \qquad (35)$$

**Case II** No Clipping $(\gamma\eta_k = \gamma\frac{\langle d^k, v^k \rangle}{4\beta^2}; \frac{\langle d^k, v^k \rangle}{4\beta^2} \le \rho)$.

$$0 \le f(x^k) - f(x^{k+1}) - \gamma\frac{\langle d^k, v^k \rangle}{4\beta^2}\langle d^k, v^k \rangle - \gamma\eta_k\langle \nabla f(x^k) - d^k, v^k \rangle + \frac{L}{2}\gamma^2\left(\frac{\langle d^k, v^k \rangle}{4\beta^2}\right)^2\|v^k\|^2$$

$$\le f(x^k) - f(x^{k+1}) - \gamma\frac{\langle d^k, v^k \rangle^2}{4\beta^2} - \gamma\eta_k\langle \nabla f(x^k) - d^k, v^k \rangle + L\gamma^2\frac{\langle d^k, v^k \rangle^2}{8\beta^2} \qquad (36)$$

$$\le f(x^k) - f(x^{k+1}) - \gamma\frac{\langle d^k, v^k \rangle^2}{8\beta^2} - \gamma\eta_k\langle \nabla f(x^k) - d^k, v^k \rangle$$

where in the last inequality we have used that $\gamma \leq \frac{1}{L}$. Rearranging,

$$
\begin{aligned}
0 &\leq f(x^k) - f(x^{k+1}) - \gamma \frac{\langle d^k, v^k \rangle^2}{8\beta^2} - \gamma \eta_k \langle \nabla f(x^k) - d^k, v^k \rangle \\
&\leq f(x^k) - f(x^{k+1}) - \frac{1}{8\beta^2} \left( \frac{\gamma}{2} \langle \nabla f(x^k), x^k - u \rangle^2 - 2\gamma \langle \nabla f(x^k) - d^k, x^k - u \rangle^2 \right) - \gamma \eta_k \langle \nabla f(x^k) - d^k, v^k \rangle \\
&\leq f(x^k) - f(x^{k+1}) - \frac{1}{8\beta^2} \left( \frac{\gamma}{2} \langle \nabla f(x^k), x^k - u \rangle^2 \right) + \frac{\gamma}{4\beta^2} \langle \nabla f(x^k) - d^k, x^k - u \rangle^2 - \gamma \eta_k \langle \nabla f(x^k) - d^k, v^k \rangle \\
&\leq f(x^k) - f(x^{k+1}) - \frac{\gamma}{16\beta^2} \langle \nabla f(x^k), x^k - u \rangle^2 + \frac{\gamma}{4\beta^2} \langle \nabla f(x^k) - d^k, x^k - u \rangle^2 - \gamma \eta_k \langle \nabla f(x^k) - d^k, v^k \rangle \\
&\leq f(x^k) - f(x^{k+1}) - \frac{\gamma}{16\beta^2} \langle \nabla f(x^k), x^k - u \rangle^2 + \frac{D_2^2 \gamma}{4\beta^2} \|\lambda^k\|_2^2 + D_2 \gamma \rho \|\lambda^k\|_2 \\
&\leq 16\beta^2 \left( f(x^k) - f(x^{k+1}) \right) - \gamma \langle \nabla f(x^k), x^k - u \rangle^2 + 4D_2^2 \gamma \|\lambda^k\|_2^2 + 16 D_2 \beta^2 \gamma \rho \|\lambda^k\|_2.
\end{aligned}
\tag{37}
$$

The rest of the proof is exactly the same as it was for Variant 1, so we omit it. $\qquad \square$

## C Experiments

Our implementations follow Unconstrained ClippedScion and ClippedScion Algorithm 3 and Algorithm 4 (Variant 2), respectively. For simplicity, we absorb the latter's factor of 4 into the clipping threshold $\rho$, so both algorithms directly clip $\sum_{l=1}^{D} \langle d_l^k, v_l^k \rangle$ at $\rho$.

CIFAR10 experiments are run on a single A100 NVIDIA GPU, NanoGPT runs are run on $4 \times$ H100 NVIDIA GPUs, and ViT experiments use $16 \times$ GH200 NVIDIA GPUs. Hyperparameters are provided in Tables 2 to 4.

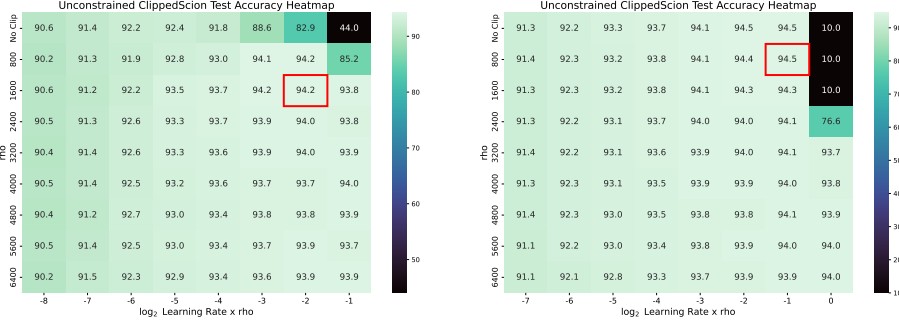

Figure 3: The optimal hyperparameters for Unconstrained ClippedScion on CIFAR10 for 80 epochs, (left) no stepsize decay (right) with stepsize decay. (indicated in red). The first row indicated with "No Clip" corresponds to Unconstrained Scion.

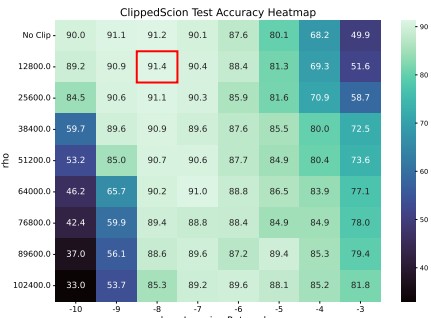

Figure 4: The optimal hyperparameters for ClippedScion on CIFAR10 for 80 epochs, no stepsize decay (indicated in red). The first row indicated with "No Clip" corresponds to Scion.

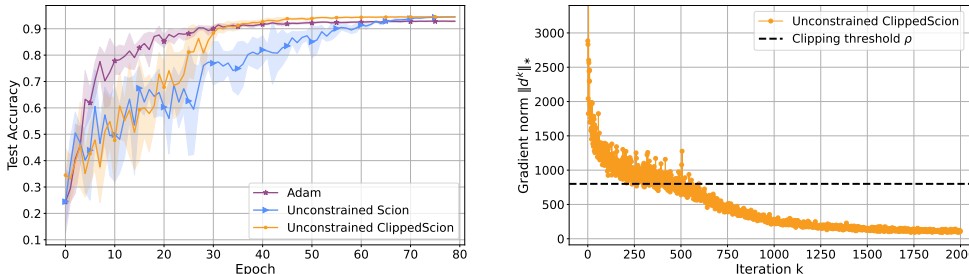

Figure 5: For CIFAR10 experiments with stepsize decay; Unconstrained Scion and Unconstrained ClippedScion achieve similar performance as expected.

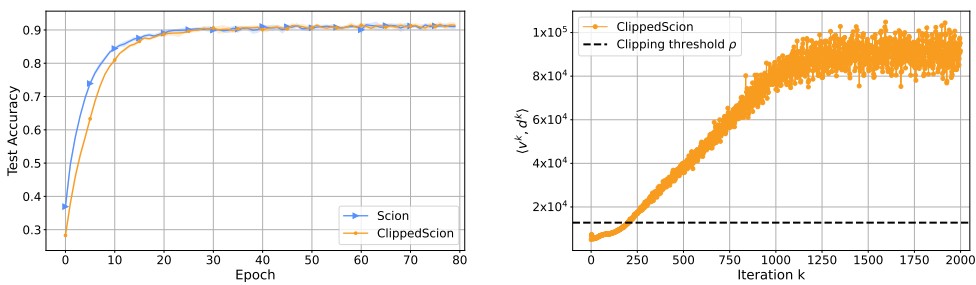

Figure 6: For CIFAR10 experiments for constrained variant of the algorithms without stepsize decay; clipping is less effective due to the surprising increase of $\langle v^k, d^k \rangle$. We observe that even the (deterministic) Wolfe gap is increasing, which is otherwise expected to go to zero.

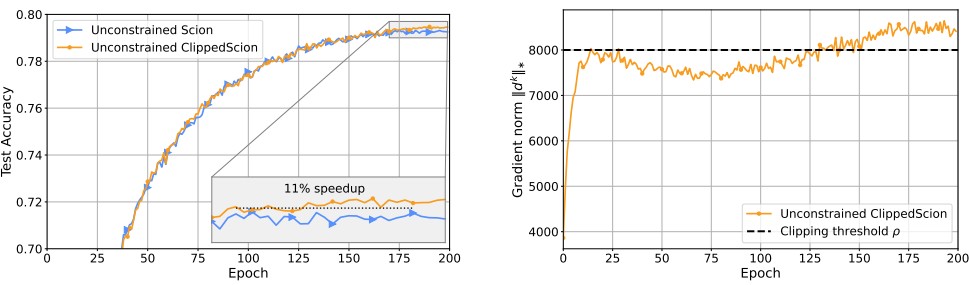

Figure 7: Clipping improves over Scion by a 11% speedup on DeiT-base. Cosine learning rate schedule is used.

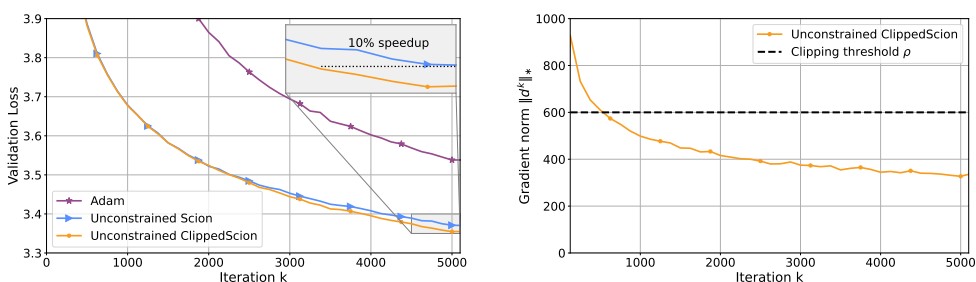

Figure 8: For fixed stepsize comparison clipping improves over Scion by more than a 10% speedup on NanoGPT (124M).

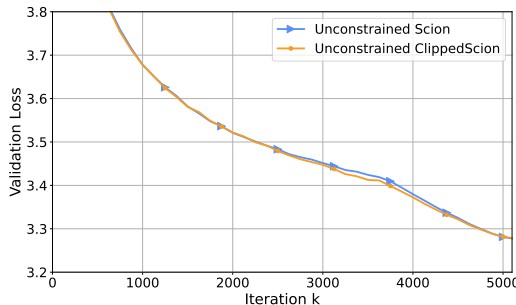

Figure 9: NanoGPT (124M) with stepsize decay. Unconstrained Scion and Unconstrained Clipped-Scion similar performance for the final iterate as expected under stepsize decay.

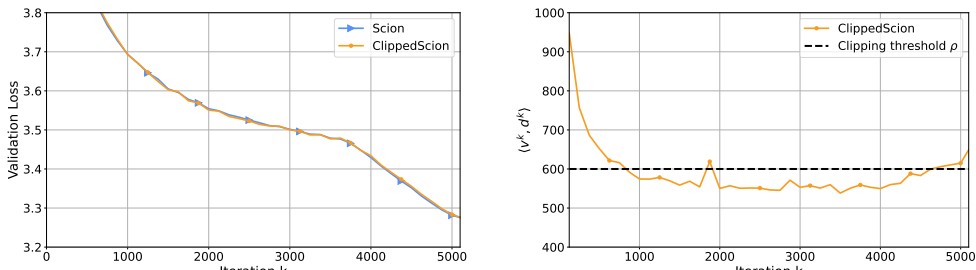

Figure 10: NanoGPT (124M) for constrained variants of the algorithm with stepsize decay. An interesting observation, which requires further investigation, is that $\langle v^k, d^k \rangle$ surprisingly increases during the linear stepsize decay.

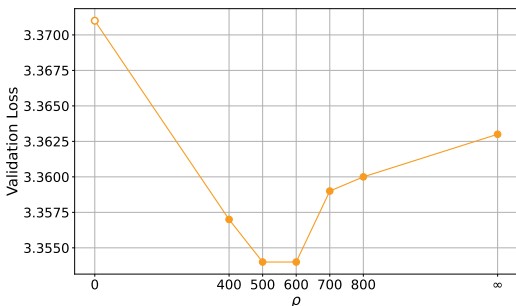

Figure 11: NanoGPT (124M) for Unconstrained ClippedScion with $\rho$ sweeping. The sweep range is set according to the gradient norm from Figure 8 (right). Both steepest descent ($\rho = \infty$) and conditional gradient ($\rho \to 0$) perform worse than clipping.

Table 2: Hyperparameters for the CIFAR10 experiments building on `airbench` [Jordan, 2024].

| Hyperparameter | Adam | (Clipped)Scion | Unconst. Scion | Unconst. ClippedScion |
|---|---|---|---|---|
| Block size (b1, b2, b3) | width factor $\times$ (64, 256, 256) | | | |
| Activation function | GELU | | | |
| Dataset | CIFAR10 (50000 training examples) | | | |
| batch size | 2000 | | | |
| Epochs | 80 | | | |
| Stepsize schedule | Linear decay $\gamma_k = \gamma \cdot (1 - k/n)$ | | | |
| Averaging parameter $\alpha$ | 0.9 | 0.5 | | |
| Stepsize $\gamma$ | 1e-3 | $2^{-8}$ | $2^{-5}$ | $2^{-2}$ |
| Initial stepsize $\gamma$ for decay | 2e-3 | - | $2^{-1}$ | $2^{-1}$ |
| Clipping parameter $\rho$ | - | 12800 | - | 1600 |
| Radius $r_1$ / $r_\ell$ / $r_D$ | - | 1 / 5 / 2000 | 1 / 5 / 200 | 1 / 5 / 200 |

Table 3: DeiT-base hyperparameters following the tuned hyperparameters of Pethick et al. [2025]

| Hyperparameter | Unconstrained Scion | Unconstrained ClippedScion |
|---|---|---|
| Layers | 12 | |
| Head dim | 64 | |
| Activation function | $\sqrt{2}\cdot$ GELU (scaled to preserve variance) | |
| Normalization function | RMSNorm | |
| Sequence Length | 197 | |
| Dataset | ImageNet-1k | |
| Stepsize schedule | Cosine decay | |
| Max lr | 0.00024 | |
| Warmup epochs | 0 | |
| End lr | $10^{-7}$ | |
| Batch size | 4096 | |
| Epochs | 200 | |
| Averaging parameter $\alpha$ | 0.1 | |
| Radius $\rho_1$ / $\rho_\ell$ / $\rho_L$ | 25 / 25 / 500 | |
| Clipping parameter $\rho$ | - | 8000 |

Table 4: NanoGPT hyperparameters following the tuned hyperparameters of Pethick et al. [2025].

| Hyperparameter | AdamW | (Unconstrained) Scion | (Unconstrained) ClippedScion |
|---|---|---|---|
| Layers | 12 | | |
| Head dim | 128 | | |
| Activation function | $2 \cdot \text{ReLU}(x)^2$ | | |
| Vocabulary size | 50304 | | |
| Dataset | FineWeb | | |
| batch size | 512 | | |
| block size | 1024 | | |
| Iterations $n$ | 5100 | | |
| Warmdown | 28.5% or 0% | | |
| Stepsize schedule | Constant then linear decay $\gamma_k = \begin{cases} \gamma & \text{if } k < n - m \\ \gamma \cdot (\frac{n-k}{m}) & \text{if } k \geq n - m \end{cases}$ | | |
| Warmup | 5% | 0 | |
| Gradient clipping | Yes | No | |
| Momentum $\beta_1$ / $\beta_2$ | 0.9 / 0.95 | - | |
| Averaging parameter $\alpha$ | - | 0.1 | |
| Stepsize $\gamma\rho$ | 0.0018 | 0.00036 | |
| Clipping parameter $\rho$ | - | 600 for 124M model and 6000 for 1B model | |
| Radius $r_1$ / $r_\ell$ / $r_D$ | - | - /50 / 3000 | |

