# OpenReview forum: "Generalized Gradient Norm Clipping & Non-Euclidean $(L_0,L_1)$-Smoothness"
_NeurIPS.cc/2025/Conference — NeurIPS 2025 oral_

### Official Review · Reviewer_cFZM · 2025-07-01

**Clarity:** 3
**Significance:** 3
**Originality:** 3
**Rating:** 5
**Confidence:** 3

**Summary:**

The paper introduces generalized gradient norm clipping (GGNC), a hybrid optimization method that combines steepest descent and conditional gradient approaches. It extends the concept of gradient norm clipping to non-Euclidean spaces and incorporates weight decay while maintaining convergence guarantees. The proposed method achieves a convergence rate of $\mathcal{O}(n^{-1/4})$ in stochastic settings. Experimental results on image classification and language modeling demonstrate that GGNC outperforms the Adam optimizer in terms of both performance and convergence speed.

**Questions:**

Q1. Can the bounded variance assumption used for the stochastic convergence results in Section 4.2 be relaxed?

Q2. Why is the generalized $(L_0, L_1)$-smoothness condition a suitable condition to study for neural networks, as claimed in Conclusion?

Q3. Can GGNC be compared with the Muon optimizer [Jordan et al., 2024b], for example, on NanoGPT?

By the way, there is a typo on Line 290, where Figure 2 should be replaced with Figure 1.

**Ethical Concerns:**

["NO or VERY MINOR ethics concerns only"]

**Final Justification:**

The responses addressed all of my concerns thoroughly.

**Limitations:**

Yes

**Quality:**

3

**Strengths And Weaknesses:**

**Strengths:**

1）The proposed GGNC effectively integrates steepest descent and conditional gradient approaches, extending gradient norm clipping to non-Euclidean settings and even constrained problems.

2）In the deterministic case, GGNC is proven to be a descent method, under the generalized $(L_0, L_1)$-smoothness condition.

3）In stochastic settings, GGNC achieves an order-optimal $\mathcal{O}(n^{-1/4})$ convergence rate by utilizing a momentum-based gradient estimator.

**Weaknesses:**

1）Although the numerical results, especially the result in Fig. 2, are interesting, more experimental results on large-scale model training (e.g., the GPT-2 model series) and the ablation study are lacking and would be valuable.

---

> ### Author Rebuttal · Authors · 2025-07-31
>
> We thank the reviewer for the positive feedback.
> The remaining questions are addressed below.
>
> > Although the numerical results, especially the result in Fig. 2, are interesting, more experimental results on large-scale model training (e.g., the GPT-2 model series) and the ablation study are lacking and would be valuable.
>
> To address the reviewer's comment, we carry out the following additional experiments:
>
> - training a NanoGPT with 1B parameters in addition to the 124M parameter model in the paper
> - training a ViT on ImageNet
> - a sensitivity study of the radius $\rho$
>
>
> **NanoGPT 1B**:
>
>
> | Model size | Method | Final val. loss | speedup | optimal lr | lr sweep |
> | -------- | -------- | -------- | -- | -- | -- |
> |  124M    | Adam     |   3.43   | | $2^{-9}$ |  $2^{-16},2^{-15},...,2^{-6}$ |
> |  124M    | Scion     | 3.37 | | $2^{-12}$ | $2^{-17},2^{-16},...,2^{-7}$ |
> |  124M    | ClippedScion    | **3.35** |  10% over Scion    | $2^{-12}$ | $2^{-17},2^{-16},...,2^{-7}$ |
> |  1B    | Adam     |  3.14   |  | $2^{-11}$ | $2^{-16},2^{-15},...,2^{-6}$ |
> |  1B    | Scion     |  3.08   | | $2^{-12}$ | $2^{-17},2^{-16},...,2^{-7}$ |
> |  1B    | ClippedScion    | **3.06** | 10% over Scion  | $2^{-12}$ | $2^{-17},2^{-16},...,2^{-7}$ |
>
> For the above NanoGPT experiments we observe that:
>
> - The optimal learning rate transfers for ClippedScion (this is not the case for Adam).
>     We make sure that the learning rate is not suboptimal for any of the methods by sweeping over a dense grid (multiples of 2) and ensuring that the best learning rate is not on the boundary of this grid.
> - Clipping leads to a 10% speedup over Scion also for the 1B model. The speedup is measuring how much faster ClippedScion can achieve the same final validation loss of Scion.
>
> **ViT on ImageNet**:
> We also run additional experiments of DeiT-base model on ImageNet following the Scion codebase, which contains a tuned Scion baseline. Specifically, we train a DeiT-base model for 200 epochs with cosine learning rate scheduling and make a reasonable guess of $\rho$ for ClippedScion. We observe that Unconstrained ClippedScion achieves 0.2% improvement over Unconstrained Scion.
> Note that this improvement is _without_ tuning $\rho$, since one run for this additional experiment already required 16xH200 GPUs for 12h.
> We will include this experiment in the final version of the paper.
>
> | Method | Test accuracy | speedup |
> |---|---|---|
> |Unconstrained Scion| 79.25% | |
> |Unconstrained ClippedScion| **79.46%** | 15% |
>
> **Sensitivity to $\rho$**:
> We conduct additional experiments to test the sensititve to $\rho$.
> The sweep range is set according to the gradient norm from Figure 2 (right). The final performance is very stable across different $\rho$.
>
> Note that the two extreme cases of steepest descent ($\rho=\infty$) and unconstrained conditional gradient ($\rho=0$) both have significantly worse performance, which provides further evidence of the advantage of the hybrid clipping method.
> We will include a plot in the final version of the paper.
>
>
> |     $\rho$   | $0$ | 400   | 500   | 600   | 700   | 800   | $\infty$ |
> | ------------ |--| ----- | ----- | ----- | ----- | ----- | -- |
> | **Val loss** | 3.371 | 3.357 | **3.354** | **3.354** | 3.359 | 3.360 | 3.363 |
>
> > Can the bounded variance assumption used for the stochastic convergence results in Section 4.2 be relaxed?
>
> Yes, we expect that it should eventually be possible to relax the standard bounded variance assumption we have used, although we have not investigated this in detail. We are aware of several works that have relaxed this assumption in other algorithms (e.g., SGD) and it seems possible that the techniques used there could be combined with our analysis of GGNC to produce some results. Further studies on this, as well as clipping the gradient before adding it to the momentum estimator (which is known to be helpful for SGD with momentum in the case of heavy-tailed noise), are certainly fruitful directions to continue this work in.
>
> > Why is the generalized $(L_0,L_1)$-smoothness condition a suitable condition to study for neural networks, as claimed in Conclusion?
>
> The best empirical evidence we are aware of is in a concurrent work on non-Euclidean $(L_0,L_1)$-smoothness [1] in which the assumption is emperically validated (see Section 5, specifically Figure 2 and Figure 4 on NanoGPT and CNN respectively).
> It seems that the assumption very closely matches the estimated Lipschitz constant between consecutive iterates (this is ultimately what is needed for the proof).
>
> [1] Artem Riabinin, Egor Shulgin, Kaja Gruntkowska, Peter Richtárik "Gluon: Making Muon & Scion Great Again! (Bridging Theory and Practice of LMO-based Optimizers for LLMs)"
>
> > Can GGNC be compared with the Muon optimizer [Jordan et al., 2024b], for example, on NanoGPT?
>
> The main mechanism used in Muon is the spectral LMO, computed using the Newton-Schulz iterations. We compare in our paper to Scion, which we believe is a strong representative of a spectral LMO based method, since it performs slightly better than Muon on NanoGPT training and since it provides a way to directly see the effect of clipping. Between Scion and Clipped Scion, the only difference is the clipped stepsize. In contrast, if we tried to compare clipped Scion to Muon, we would have to account for the fact that Muon uses AdamW on the vector parameters and a different momentum, which would hinder interpretation.
>
> > By the way, there is a typo on Line 290, where Figure 2 should be replaced with Figure 1.
>
> Thanks for catching the typo, we will correct it.

---

> > ### Comment · Reviewer_cFZM · 2025-08-04
> >
> > Thanks. The responses addressed all of my concerns thoroughly.

---

### Official Review · Reviewer_wW69 · 2025-07-03

**Clarity:** 3
**Significance:** 2
**Originality:** 2
**Rating:** 2
**Confidence:** 3

**Summary:**

This paper introduces a training algorithm that combines norm-constrained linear minimization oracles (LMOs) with gradient clipping. The authors assume $(L_0,L_1)$-smoothness in their theoretical framework and prove an optimal convergence rate of $O(n^{-1/4})$. The performance of their algorithm is demonstrated through experiments on CIFAR-10 and NanoGPT.

**Questions:**

1. Is the $(L_0,L_1)$-smoothness condition valid in non-Euclidean settings? Do you have empirical verification of this assumption?
2. Compared to Pethick et al. (2025), the optimal convergence rate remains the same in the stochastic case. Why is there no improvement despite additional smoothness assumptions?
3. Pethick et al. (2025) use LMOs to avoid computing the dual norm, but your paper requires dual norm computation for gradient norm clipping. Is this why your learning rate depends on the Lipschitz constant while Pethick et al. (2025) does not?
4. When obtaining $\text{lmo}(d)$, $x$ is already constrained within domain $D$. Why not simply control the norm-ball $D$ directly?

**Ethical Concerns:**

["NO or VERY MINOR ethics concerns only"]

**Final Justification:**

I believe both the theoretical and empirical contributions of this paper are incremental, as explained in my earlier comments. Therefore, I tend to keep my score unchanged.

**Limitations:**

The theoretical contribution appears incremental, and there is no significant improvement in empirical performance.

**Paper Formatting Concerns:**

I did not notice any major formatting issues in the paper.

**Quality:**

3

**Strengths And Weaknesses:**

Strengths:
1. The paper is well-written with clearly presented theoretical results.
2. The authors extend the understanding of clipping benefits (previously limited to the Euclidean setting) to arbitrary norms, proving descent under a novel non-Euclidean $(L_0,L_1)$-smoothness condition.

Weaknesses:
1. The motivation for using gradient norm clipping is inadequately explained. The approach does not appear to improve the convergence rate.
2. The empirical performance of the proposed algorithm appears nearly identical to the unclipped version.

---

> ### Author Rebuttal · Authors · 2025-07-31
>
> We thank the reviewer for the feedback and address the remaining concerns and questions below.
>
> > The empirical performance of the proposed algorithm appears nearly identical to the unclipped version.
>
> We politely disagree. The difference in validation loss translates into a 10% speedup for NanoGPT, which we consider substantial.
> We additionally carry out the following experiments in order to provide further evidence:
>
> - training a NanoGPT with 1B parameters in addition to the 124M parameter model in the paper
> - training a ViT on ImageNet
> - sensitivity of $\rho$
>
> **NanoGPT 1B**:
>
>
> | Model size | Method | Final val. loss | speedup | optimal lr | lr sweep |
> | -------- | -------- | -------- | -- | -- | -- |
> |  124M    | Adam     |   3.43   | | $2^{-9}$ |  $2^{-16},2^{-15},...,2^{-6}$ |
> |  124M    | Scion     | 3.37 | | $2^{-12}$ | $2^{-17},2^{-16},...,2^{-7}$ |
> |  124M    | ClippedScion    | **3.35** |  10% over Scion    | $2^{-12}$ | $2^{-17},2^{-16},...,2^{-7}$ |
> |  1B    | Adam     |  3.14   |  | $2^{-11}$ | $2^{-16},2^{-15},...,2^{-6}$ |
> |  1B    | Scion     |  3.08   | | $2^{-12}$ | $2^{-17},2^{-16},...,2^{-7}$ |
> |  1B    | ClippedScion    | **3.06** | 10% over Scion  | $2^{-12}$ | $2^{-17},2^{-16},...,2^{-7}$ |
>
> For the above NanoGPT experiments we observe that:
>
> - The optimal learning rate transfers for ClippedScion (this is not the case for Adam).
>     We make sure that the learning rate is not suboptimal for any of the methods by sweeping over a dense grid (multiples of 2) and ensuring that the best learning rate is not on the boundary of this grid.
> - Clipping leads to a 10% speedup over Scion also for the 1B model. The speedup is measuring how much faster ClippedScion can achieve the same final validation loss of Scion.
>
> **ViT on ImageNet**:
> We also run additional experiments of DeiT-base model on ImageNet following the Scion codebase, which contains a tuned Scion baseline. Specifically, we train a DeiT-base model for 200 epochs with cosine learning rate scheduling and make a reasonable guess of $\rho$ for ClippedScion. We observe that Unconstrained ClippedScion achieves 0.2% improvement over Unconstrained Scion.
> Note that this improvement is _without_ tuning $\rho$, since one run for this additional experiment already required 16xH200 GPUs for 12h.
> We will include this experiment in the final version of the paper.
>
> | Method | Test accuracy | speedup |
> |---|---|---|
> |Unconstrained Scion| 79.25% | |
> |Unconstrained ClippedScion| **79.46%** | 15% |
>
> **Sensitivity to $\rho$**:
> We conduct additional experiments to test the sensitivity to $\rho$.
> The sweep range is set according to the gradient norm from Figure 2 (right). The final performance is very stable across different $\rho$.
>
> Note that the two extreme cases of steepest descent ($\rho=\infty$) and unconstrained conditional gradient ($\rho=0$) both have significantly worse performance, which provides further evidence of the advantage of the hybrid clipping method.
> We will include a plot in the final version of the paper.
>
>
> |     $\rho$   | $0$ | 400   | 500   | 600   | 700   | 800   | $\infty$ |
> | ------------ |--| ----- | ----- | ----- | ----- | ----- | -- |
> | **Val loss** | 3.371 | 3.357 | **3.354** | **3.354** | 3.359 | 3.360 | 3.363 |
>
>
> > The motivation for using gradient norm clipping is inadequately explained. The approach does not appear to improve the convergence rate.
>
> Indeed, the convergence rate we find is matching the one given in [1]. Ultimately, this is a worst-case convergence rate and, in practice, the behavior of the two algorithms is not always the same, even if there exists some pathological function for which both algorithms would have the same rate of convergence. This is evidenced by our accompanying numerical experiments, which confirm the benefits of clipping under constant learning rate in a practical setting like training a NanoGPT network.
>
> [1] Thomas Pethick, Wanyun Xie, Kimon Antonakopoulos, Zhenyu Zhu, Antonio Silveti-Falls and Volkan Cevher, "Training Deep Learning Models with Norm-Constrained LMOs"
>
> > Is the $(L_0, L_1)$-smoothness condition valid in non-Euclidean settings? Do you have empirical verification of this assumption?
>
> Yes, there is evidence to support that.
> In a concurrent work on non-Euclidean $(L_0,L_1)$-smoothness [1], the assumption is empirically validated (see Section 5, specifically Figure 2 and Figure 4 on NanoGPT and CNN respectively).
> It seems that the assumption very closely matches the estimated Lipschitz constant between consecutive iterates (this is ultimately what is needed for the proof).
>
> [1] Artem Riabinin, Egor Shulgin, Kaja Gruntkowska, Peter Richtárik "Gluon: Making Muon & Scion Great Again! (Bridging Theory and Practice of LMO-based Optimizers for LLMs)"
>
> > Compared to Pethick et al. (2025), the optimal convergence rate remains the same in the stochastic case. Why is there no improvement despite additional smoothness assumptions?
>
> The $(L_0,L_1)$-smoothness assumption is a strictly _weaker_ assumption (i.e., more general, thus allowing for a broader class of functions).
>
> > Pethick et al. (2025) use LMOs to avoid computing the dual norm, but your paper requires dual norm computation for gradient norm clipping. Is this why your learning rate depends on the Lipschitz constant while Pethick et al. (2025) does not?
>
> Yes ClippedScion needs to set the stepsize clipping factor $\rho$, but can then converge faster in practice as seen in the experiments.
>
> > When obtaining $\mathrm{lmo}(d)$, $x$ is already constrained within domain $\mathcal{D}$. Why not simply control the norm-ball $\mathcal{D}$ directly?
>
> Decreasing the radius is in some sense what Scion does when the stepsize is taken diminishing ($\rho$ and $\gamma$ plays the same role in the unconstrained case as e.g. seen in Theorem 4.5).

---

> > ### Comment · Reviewer_wW69 · 2025-08-03
> >
> > I thank the authors for the additional explanations and experiments. However, the effect of gradient norm clipping appears limited to me. The authors claim a 10% speed-up, but looking at the curves in Figure 2, there seems to be almost no visible difference, and it is unclear whether such a speed-up would be noticeable in practice.
> >
> > Second, the convergence analysis does not improve the convergence rate, and there are no new techniques introduced in the analysis, so in my view, the theoretical contribution remains limited.
> >
> > Therefore, I tend to keep my score unchanged.
> >
> > Additionally, [1] empirically verifies layer-wise smoothness. Does this imply that your assumption holds for arbitrary norms?
> >
> > [1] Artem Riabinin, Egor Shulgin, Kaja Gruntkowska, Peter Richtárik. Gluon: Making Muon & Scion Great Again! (Bridging Theory and Practice of LMO-based Optimizers for LLMs)

---

> > > ### Author Response · Authors · 2025-08-05
> > > **Response to Reviewer wW69**
> > >
> > > We thank the reviewer for taking the time to respond to our rebuttal.
> > >
> > > > I thank the authors for the additional explanations and experiments. However, the effect of gradient norm clipping appears limited to me. The authors claim a 10% speed-up, but looking at the curves in Figure 2, there seems to be almost no visible difference, and it is unclear whether such a speed-up would be noticeable in practice.
> > >
> > > The plot in Figure 2 is showing the perplexity, which appears quite similar for both curves due to the scaling of the plot. However, the speed-up we claim is measured rigorously based on the runs of the algorithm. We believe that measuring the time-to-result, e.g., measuring how quickly an algorithm can reach a given validation loss (result), is an accepted way to compare the performance of optimizers. Time-to-result, is for instance, what is used in the celebrated AlgoPerf benchmark by Google.
> > >
> > > > Second, the convergence analysis does not improve the convergence rate, and there are no new techniques introduced in the analysis, so in my view, the theoretical contribution remains limited.
> > >
> > > We believe there is a misunderstanding:
> > >
> > > - We are, first and foremost, not trying to improve the rates. Rather, we are trying to relax the Lipschitz conditions, which we do for both conditional gradient methods (Theorem 4.5) and our hybrid method (Theorem 4.3). Theorem 4.3 and Theorem 4.5 show a clear separation between these two methods for fixed stepsizes: our hybrid method converges due to the descent property, whereas the conditional gradient method does not converge (only to a neighborhood governed by the stepsize). Fixed stepsize is becoming increasingly popular due to its simplicity - see e.g., [2].
> > > - Regarding novelty, there are certainly new techniques introduced. For instance, because of the interplay between clipping and the momentum estimator we need to deal with problematic quantities such as $\|d^k\|_{\ast}\|\nabla f(x^k)\|_{\ast}$, which do not otherwise appear in previous work (see e.g., case II in the proof of Theorem 4.9).
> > >    Furthermore, to show convergence for the short step in the stochastic case, it is _necessary_ to generalize the stepsize as done with Algorithm 2 – specifically the introduction of the parameter $\rho$ is essential for establishing converges and the typical analysis for the short step would not suffice.
> > >
> > > We insist that these are bona fide theoretical innovations.
> > >
> > >
> > > [2] ByteDance Seed. "Model Merging in Pre-training of Large Language Models" 2025
> > >
> > >
> > > > Additionally, [1] empirically verifies layer-wise smoothness. Does this imply that your assumption holds for arbitrary norms?
> > > > [1] Artem Riabinin, Egor Shulgin, Kaja Gruntkowska, Peter Richtárik. Gluon: Making Muon & Scion Great Again! (Bridging Theory and Practice of LMO-based Optimizers for LLMs)
> > >
> > > It is important to point out that [1] validates the assumption for _exactly_ the same norm choice as in our work, since we both build on the Scion optimizer, which makes a norm choice for the entire neural network based on layerwise norms. Specifically, we both choose the $\ell_\infty$-norm for the last layer and spectral norm for the remaining layers (see their Figure 8).
> > >
> > > Regarding whether the assumption holds for arbitrary norms:
> > > Since the problem is posed in a finite-dimensional vector space, all norms are technically equivalent (up to some constant factor) and so empirically verifying the layer-wise smoothness assumption for one norm is sufficient to guarantee layer-wise smoothness for arbitrary norms.

---

### Official Review · Reviewer_cpYw · 2025-07-03

**Clarity:** 3
**Significance:** 3
**Originality:** 3
**Rating:** 5
**Confidence:** 3

**Summary:**

This paper proposes a new technique that generalizes gradient clipping to non-Euclidean ($\ell_2$) norms. On the non-Euclidean norms, standard gradient descent is generalized to steepest descent with update $d^\\# = \mathrm{argmax}\_x \langle d,x\rangle - \frac{1}{2}\\|x\\|^2$, while normalized gradient is generalized to conditional gradient via linear minimization oracle (lmo), that is $lmo(d) = \mathrm{argmin}\_{\\|x\\|=1} \langle d,x\rangle$. However, it was unclear what is the correct analogy of gradient clipping on non-Euclidean norms. To close this gap, this paper observes a key connection between the two methods, that is $lmo(d) = -d^\\# / \\|d\\|\_* $, and proposes the generalized gradient clipping defined as $\min\\{1, \frac{\rho}{\\|d\\|\_*} \\} \cdot d^\\#$.

Based on this core idea, this paper provides two variants of gradient clipping, for unconstrained and constrained problems. In both settings, the authors provided convergence analysis in both deterministic and stochastic setting, and proved that the proposed gradient clipping algorithm converges to the optimal rate. In addition, the authors also validated the algorithm on both image classification and nanoGPT pre-training tasks, and showed that the proposed algorithm achieves better performance compared to the Adam baseline and Scion, an algorithm based on conditional gradient (analogous to normalized gradient on carefully chosen norms) instead of gradient clipping. These empirical results demonstrate the practical benefits of the new algorithm.

**Questions:**

NA

**Ethical Concerns:**

["NO or VERY MINOR ethics concerns only"]

**Final Justification:**

Sorry for the late response, I wasn't aware of the new requirement of this year. I have read the responses from the authors and the comments from other reviewers, and I keep my current score as the recommended score.

**Limitations:**

yes

**Quality:**

3

**Strengths And Weaknesses:**

This paper has several notable strengths. First of all, it introduces a novel technique that extends gradient clipping to non-Euclidean norms. By connecting steepest descent update and lmo update in conditional gradient, the authors managed to define a meaningful clipping on general norms.

Additionally, the paper makes a solid contribution to the literature by addressing the previously unresolved question of how to implement gradient clipping for general non-Euclidean norms. Prior to this work, the correct analogy for gradient clipping in non-Euclidean norms was unclear, leaving a gap in the understanding of optimization techniques. The authors successfully fill this gap, providing a valuable resource for researchers and practitioners looking to improve their optimization strategies.

Finally, the proposed algorithm is supported by solid evidences, both in theoretical analysis and empirical experiments. The authors demonstrate that their proposed gradient clipping algorithm converges to the optimal rate in both deterministic and stochastic settings, and for both constrained and unconstrained problems. This theoretical foundation is complemented by empirical validation on tasks of image classification and nanoGPT pre-training, where the algorithm outperforms existing methods like the Adam baseline and Scion. These results demonstrate the empirical effectiveness of gradient clipping with general norms.

Overall, I think this paper has solid results, thus recommend acceptance.

---

> ### Author Rebuttal · Authors · 2025-07-31
>
> Thank you for the positive review - we remain available if there are any further questions.

---

> > ### Comment · Reviewer_cpYw · 2025-08-07
> >
> > Thank you. I don't have further questions and remain my score.

---

### Official Review · Reviewer_eHBQ · 2025-07-03

**Clarity:** 2
**Significance:** 2
**Originality:** 2
**Rating:** 4
**Confidence:** 4

**Summary:**

The paper proposes Generalized Gradient Norm Clipping (GGNC), a hybrid first-order method that interpolates between steepest descent (SD) and an unconstrained conditional-gradient (uCG) step by clipping the dual gradient norm with respect to an arbitrary norm. Under a generalized (L₀, L₁)-smoothness assumption, the authors prove:

1. A descent lemma and an O(n⁻¹) deterministic rate to a neighborhood followed by O(n⁻¹/²) convergence near stationary points.
2. An order-optimal O(n⁻¹/⁴) stochastic rate via a momentum estimator.
3. An interpretation of weight decay as a clipped Frank-Wolfe "short step," along with the Stochastic Short-Step Conditional Gradient (S3CG) algorithm.
4. Instantiations for ℓ∞ ("Clipped Sign"), spectral, and product norms, accompanied by small-scale experiments on CIFAR-10 and NanoGPT, suggesting faster early-stage progress compared to unclipped methods.

**Questions:**

1. Computational overhead: For spectral or product norms, the lmo requires an SVD per update; how does GGNC scale to ImageNet-size models? Please provide wall-clock comparisons.

2. Sensitivity to ρ: Figures use a fixed clipping threshold, but how robust is performance to mis-specifying ρ by ±2×? A sweep would strengthen claims.

3. Adaptive stepsizes: Your theory requires γ ≤ 1/L₀. In practice, L₀ is unknown. Can backtracking or AdaGrad be combined with GGNC while retaining descent? Empirical evidence would increase the quality score.

4. Baselines: Why are AdamW, Lion, and normalized SGD with momentum absent? Demonstrating advantages over these would affect my significance assessment.

5. Assumption tightness: Is (L₀, L₁) actually satisfied for deep nets in the modular norm? Please supply empirical Lipschitz estimates or counterexamples.

I would consider increasing my score if robust large-scale experiments with adaptive stepsizes and stronger baselines show clear benefits, or if you provide a cheap approximation of the lmo/sharp operator that preserves convergence.

**Ethical Concerns:**

["NO or VERY MINOR ethics concerns only"]

**Final Justification:**

I thank the authors for their rebuttal, which has satisfactorily addressed my concerns. I'm increasing my score.

**Limitations:**

The paper includes a "Limitations" checklist entry but does not discuss computational cost, hyperparameter sensitivity, or failure modes in depth. These should be added.

**Quality:**

2

**Strengths And Weaknesses:**

Quality

Strengths:

* Rigorous extension of the (L₀, L₁) framework to arbitrary norms; proofs are mostly complete and technically competent.
* The connection between clipping and Frank-Wolfe short steps is elegant and may interest the optimization community.

Weaknesses:

* Theory largely recycles existing proof templates for Euclidean clipping; novelty is modest.
* Crucial practical aspects—computational cost of the sharp operator and lmo for deep networks, choice of radius ρ, and stability when the dual norm is ill-conditioned—are not analyzed.
* Stochastic guarantees require known L₀ and L₁ to set stepsizes, conflicting with the "parameter-agnostic" narrative in the introduction.

Clarity

Strengths:

* High-level motivation is clear and algorithms are presented in pseudocode (Algorithms 1–2).

Weaknesses:

* Notation is heavy, and several definitions (e.g., "product norm" radii) are buried mid-paragraph, making reproduction difficult.
* Experimental section omits many details (network width, regularization, hardware, etc.); Appendix C only lists a subset.
* Figures 1–2 plot clipped and unclipped variants but omit error bars for Adam, hindering interpretation.

Significance

Strengths:

* May stimulate further work on non-Euclidean clipping and constrained optimization links.

Weaknesses:

* Empirical impact is unconvincing: gains on CIFAR-10 are approximately 1–2 percentage points and vanish once a learning-rate schedule is introduced.
* Experiments are confined to toy-scale models; no evidence on large-scale vision or language tasks where clipping is commonly used.
* Competing adaptive methods (AdamW, Lion, RMSProp) are not compared, limiting relevance for practitioners.

Originality

Strengths:

* The specific synthesis of SD, uCG, and Frank-Wolfe short steps under (L₀, L₁)-smoothness appears new.

Weaknesses:

* Core idea—clip the gradient in a dual norm—has been folklore; the paper formalizes but does not fundamentally change practice.
* Prior concurrent works (Gorbunov et al., Vankov et al.) already extend clipping under relaxed smoothness; the non-Euclidean lift is incremental.

---

> ### Author Rebuttal · Authors · 2025-07-31
>
> We thank the reviewer for the thorough feedback and address the remaining concerns and questions below.
>
> > I would consider increasing my score if robust large-scale experiments with adaptive stepsizes and stronger baselines show clear benefits, or if you provide a cheap approximation of the lmo/sharp operator that preserves convergence.
>
> **Large scale experiments**
> Regarding large scale experiments, the 10% speedup in Figure 2 is observed on a 124M parameter model (we will add the missing parameter count to the paper).
> We ran additional experiments to check if the conclusion is robust to even larger model sizes, specifically a 1 billion parameter model:
>
> | Model size | Method | Final val. loss | speedup | optimal lr | lr sweep |
> |--|--|--|--|--|--|
> |124M | Adam| 3.43 | | $2^{-9}$ |$2^{-16},2^{-15},...,2^{-6}$ |
> |124M | Scion| 3.37 | | $2^{-12}$ | $2^{-17},2^{-16},...,2^{-7}$ |
> |124M | ClippedScion | **3.35** |10% over Scion | $2^{-12}$ | $2^{-17},2^{-16},...,2^{-7}$ |
> |1B | Adam|3.14 || $2^{-11}$ | $2^{-16},2^{-15},...,2^{-6}$ |
> |1B | Scion|3.08 | | $2^{-12}$ | $2^{-17},2^{-16},...,2^{-7}$ |
> |1B | ClippedScion | **3.06** | 10% over Scion| $2^{-12}$ | $2^{-17},2^{-16},...,2^{-7}$ |
>
> For the above NanoGPT experiments we observe that:
>
> - The optimal learning rate transfers for ClippedScion (this is not the case for Adam).
>     We make sure that the learning rate is not suboptimal for any of the methods by sweeping over a dense grid (multiples of 2) and ensuring that the best learning rate is not on the boundary of this grid.
> - Clipping leads to a 10% speedup over Scion also for the 1B model. The speedup is measuring how much faster ClippedScion can achieve the same final validation loss of Scion.
>
> We also run additional experiments of DeiT-base model on ImageNet following the Scion codebase, which contains a tuned Scion baseline. Specifically, we train a DeiT-base model for 200 epochs with cosine learning rate scheduling and make a reasonable guess of $\rho$ for ClippedScion. We observe that Unconstrained ClippedScion achieves 0.2% improvement over Unconstrained Scion.
> Note that this improvement is _without_ tuning $\rho$, since one run for this additional experiment already required 16xH200 GPUs for 12h.
> We will include this experiment in the final version of the paper.
>
> | Method | Test accuracy | speedup |
> |---|---|---|
> |Unconstrained Scion| 79.25% | |
> |Unconstrained ClippedScion| **79.46%** | 15% |
>
>
> **Cost of LMO computation/wall-clock time**
> Regarding the computation of the lmo/sharp operator, the only relevant norm-choice for which these objects cannot be computed in closed form is the spectral norm.
> Fortunately, we can efficiently approximate the spectral LMO through a fast implementation of the Newton-Schultz iterations (see the experimental section). In practice, the Newton-Schultz iterations add a ~0.7% overhead, e.g., for NanoGPT (see [1], which our experiments are also in agreement with). This already small overhead shrinks proportionally with increasing bathsize and depth of the model.
>
> Compared with the speedup over Adam, the overhead is negligible. The wall-clock time speedup of Scion over Adam for NanoGPT in Figure 2 is 34%, while for ClippedScion it is 39%. We will include wall-clock time in the final version.
>
> [1] Keller Jordan, Yuchen Jin, Vlado Boza, Jiacheng You, Franz Cesista, Laker Newhouse and Jeremy Bernstein, "Muon: An optimizer for hidden layers in neural networks"
>
> > Sensitivity to ρ: Figures use a fixed clipping threshold, but how robust is performance to mis-specifying ρ by ±2×? A sweep would strengthen claims.
>
> We conduct additional experiments to test the sensititve to $\rho$.
> The sweep range is set according to the gradient norm from Figure 2 (right). The final performance is very stable across different $\rho$.
>
> Note that the two extreme cases of steepest descent ($\rho=\infty$) and unconstrained conditional gradient ($\rho=0$) both have significantly worse performance, which provides further evidence of the advantage of the hybrid clipping method.
> We will include a plot in the final version of the paper.
>
> |     $\rho$   | $0$ | 400   | 500   | 600   | 700   | 800   | $\infty$ |
> | -- |--| ----- | ----- | ----- | ----- | ----- | -- |
> | **Val loss** | 3.371 | 3.357 | **3.354** | **3.354** | 3.359 | 3.360 | 3.363 |
>
>
> > Notation is heavy, and several definitions (e.g., "product norm" radii) are buried mid-paragraph, making reproduction difficult.
>
> For reproduction we have summerized the full algorithm in the appendix (Algorithm 3 and 4), which includes the per layer radii explicitly.
> This should provide an overview of what steps are needed for the implementation.
> We didn't include it in the main text due to space limitations, but we will consider pulling it up.
>
> The supplementary also includes a compact pytorch implementation, which should make it straightforward to reproduce.
>
> > Experimental section omits many details (network width, regularization, hardware, etc.); Appendix C only lists a subset.
>
> - The network width is indeed missing from the hyperparameter table, thanks for catching that.
> The NanoGPT width is 768 which makes it a 124 million parameter model, which is the default in the NanoGPT repository that we build on.
> - The cifar10 experiment uses the default data augmentation from the airbench repository that we build on. We will add a description to the paper.
> - The hardware used is specified in beginning of Appendix C.
>
> > Theory largely recycles existing proof templates for Euclidean clipping; novelty is modest.
>
> We respectfully disagree with the framing of building on prior work as recycling existing proof templates. The way we are controlling the magnitude of the update is completely different to Euclidean clipping alone and so is the analysis that follows; prior works did not use non-Euclidean LMOs (e.g., spectral LMO) in combination with a clipped step size before our work, nor was it common in practice.
>
> > Prior concurrent works (Gorbunov et al., Vankov et al.) already extend clipping under relaxed smoothness; the non-Euclidean lift is incremental.
> > Core idea—clip the gradient in a dual norm—has been folklore; the paper formalizes but does not fundamentally change practice.
>
> Our change to a non-Euclidean setting goes much further than simply changing the norm in the clipping of the stepsize. The algorithm we are considering fundamentally alters the direction (and not just the magnitude) used to update the iterates through the use of the non-Euclidean LMO (a kind of nonlinear preconditioner). This significantly changes the algorithm, both in practice (we do not only clip the stepsize, we must also apply an LMO, e.g., Newton-Schulz iterations) and in its theoretical analysis (e.g., the direction of steepest descent used in the algorithm may not be unique, no inner product identities to rely on, etc).
>
> > Crucial practical aspects—computational cost of the sharp operator and lmo for deep networks, choice of radius ρ, and stability when the dual norm is ill-conditioned—are not analyzed.
>
> We have elaborated on the cost of the sharp operator and LMO for deep networks in our previous responses but, to be clear, there is very little overhead coming from the computation of the sharp operator and LMO. Regarding stability when the dual norm is ill-conditioned, it's not clear what is meant here. There is no requirement on the dual norm besides needing the LMO to be accessible (either in closed-form or approximately using a subroutine like Newton-Schulz).
>
> > Assumption tightness: Is (L₀, L₁) actually satisfied for deep nets in the modular norm? Please supply empirical Lipschitz estimates or counterexamples.
>
> Yes, there is evidence to support that.
> In a concurrent work on non-Euclidean $(L_0,L_1)$-smoothness [2], the assumption is empirically validated (see Section 5, specifically Figure 2 and Figure 4 on NanoGPT and CNN respectively).
> It seems that the assumption very closely matches the estimated Lipschitz constant between consecutive iterates (this is ultimately what is needed for the proof).
>
> [2] Artem Riabinin, Egor Shulgin, Kaja Gruntkowska, Peter Richtárik "Gluon: Making Muon & Scion Great Again! (Bridging Theory and Practice of LMO-based Optimizers for LLMs)"
>
> > Baselines: Why are AdamW, Lion, and normalized SGD with momentum absent? Demonstrating advantages over these would affect my significance assessment.
>
> AdamW has been tested in this setting and was found to be worse than Adam. Therefore we do compare against Adam but not AdamW, which would be redundant for the modded-nanoGPT repository. We focus on comparing against one strong, well-tuned baseline rather than having many baselines:
> - The Adam optimizer is tuned (across 12 runs) and warmup is added to the learning rate schedule (in contrast to Scion/ClippedScion), which significantly boosts performance.
> - In addition to comparing to Adam, we compare against Scion since it can achieve state-of-the-art on NanoGPT experiments and since it provides a direct comparison to see the effect of clipping (no other changes are made simultaneously to the optimizer).
>
> > Adaptive stepsizes: Your theory requires γ ≤ 1/L₀. In practice, L₀ is unknown. Can backtracking or AdaGrad be combined with GGNC while retaining descent? Empirical evidence would increase the quality score.
>
> We believe this is an interesting direction that we also mention in the conclusion, but it does not seem completely obvious how to generalize beyond the Euclidean case (despite our preliminary efforts). Besides this, the results we present are nonetheless novel (there were no previous results in this setting, even assuming knowledge of the constants) which is why we have left it for future work.
>
> > The paper includes a "Limitations" checklist entry but does not discuss computational cost, hyperparameter sensitivity, or failure modes in depth. These should be added.
>
> We elaborate on the limitations in the checklist, thanks for pointing this out.

---

> ### Comment · Reviewer_eHBQ · 2025-08-08
>
> I thank the authors for their rebuttal, which has satisfactorily addressed my concerns. I'm increasing my score.

---

### Comment · Area_Chair_NE6E · 2025-08-05

Dear Reviewers,

Thank you again for your time and efforts in reviewing papers for NeurIPS 2025.

I am writing to remind you that **active participation in the author-reviewer discussion phase is mandatory**. According to the guidelines from the NeurIPS program chairs, reviewers are **required to engage directly with the authors in the discussion thread**, especially in response to their rebuttals.

Please note the following important policy:

- Simply reading the rebuttal or internally considering it is **not sufficient** -- reviewers must **post at least one message to the authors**, even if it is only to confirm that their concerns were resolved. If they have not been addressed, please explain why.

- **Acknowledging the rebuttal without any engagement with the authors will be considered insufficient**. I am obligated to flag such cases using the *InsufficientReview* mechanism, which may **impact future reviewing invitations and result in desk rejection of your own submissions**.

If you have not yet responded to the authors in the discussion thread, I kindly ask you to do so **as soon as possible**, and **no later than August 8, 11:59pm AoE**.

Please don't hesitate to reach out to me if you have any questions or concerns.

Best regards,

AC

---

### Decision · Program_Chairs · 2025-09-17

**Decision:**

Accept (oral)

**Comment:**

**Summary.** This paper introduces Generalized Gradient Norm Clipping (GGNC), which extends gradient descent with Euclidean norm clipping to the non-Euclidean setting. The method interpolates between steepest descent and conditional gradient approaches. Under a newly defined non-Euclidean version of $(L_0, L_1)$-smoothness, the authors establish new convergence guarantees for GGNC in the non-convex case, generalizing prior results of Koloskova et al. (2023). The analysis is further extended to stochastic problems with bounded-variance gradient noise. In addition, the authors propose the Stochastic Short Step Conditional Gradient (S3CG) method, which combines clipping with weight decay and achieves the same convergence rate in the stochastic regime. Numerical experiments evaluating specific instances of GGNC and S3CG (Unconstrained Clipped Scion and Clipped Scion) demonstrate promising performance.

**Strengths.** The paper tackles an important problem: extending convergence analysis under generalized smoothness to non-Euclidean norms. This direction is strongly motivated by the recent success of LMO-based optimizers such as Muon and Scion in large-scale language model training. The contributions are therefore both theoretically and practically relevant. Furthermore, the work clarifies and unifies connections between different optimization methods, which is valuable for future research.

**Weaknesses.** The reviewers highlighted the following limitations: (1) modest empirical gains, (2) reliance on known proof techniques, (3) stochastic results requiring knowledge of $L_0$ and $L_1$, and (4) lack of clear motivation. The authors responded effectively with additional experiments and clarifications. In my view, point (2) is not a valid weakness - most theoretical works necessarily build on existing techniques, and novelty should not be reduced to technical tricks alone. Similarly, point (3) is a common assumption in related literature and acceptable here given this is the first work addressing the stated problem.

**Reviewers' consensus.** Reviewers cpYw, eHBQ, and cFZM expressed strong support (scores 5, 4, 5) and were satisfied with the rebuttal. Reviewer wW69 recommended rejection (score 2), raising concerns about novelty and significance in their review and during the internal AC-Reviewers discussion. I find these objections largely subjective. For example,
- Claiming the extension from the $\ell_2$ ball to the $\ell_p$ ball is “straightforward” is factually inaccurate since the authors extend to arbitrary norms, not merely $\ell_p$.
- Suggesting that $(L_0, L_1)$-smoothness reduces to standard smoothness because of clipping is incorrect, since the definition depends on the unclipped gradient.
- Questioning the motivation for clipping is subjective; in practice, clipping is a standard component of large-scale training, which itself justifies this study.

Overall, I do not find these concerns sufficient to outweigh the technical contributions and importance of the work.

**Final recommendation.** Considering the theoretical significance, the novelty of extending convergence guarantees to the non-Euclidean setting, and the positive reception from the majority of reviewers, I recommend **acceptance as an oral presentation**. The paper addresses an important and timely problem in optimization for machine learning, offers technically sound contributions, and presents encouraging empirical validation.